# Correlational networking guides the discovery of unclustered lanthipeptide protease-encoding genes

Dan Xue[1,9], Ethan A. Older[1,9], Zheng Zhong [2,3,9], Zhuo Shang[1], Nanzhu Chen[2,3], Nolan Dittenhauser[1], Lukuan Hou[1], Peiyan Cai[2,3], Michael D. Walla[4], Shi-Hui Dong [5], Xiaoyu Tang [6], Hexin Chen [7], Prakash Nagarkatti[8], Mitzi Nagarkatti[8], Yong-Xin Li [2,3✉] & Jie Li [1✉]

Bacterial natural product biosynthetic genes, canonically clustered, have been increasingly found to rely on hidden enzymes encoded elsewhere in the genome for completion of biosynthesis. The study and application of lanthipeptides are frequently hindered by unclustered protease genes required for final maturation. Here, we establish a global correlation network bridging the gap between lanthipeptide precursors and hidden proteases. Applying our analysis to 161,954 bacterial genomes, we establish 5209 correlations between precursors and hidden proteases, with 91 prioritized. We use network predictions and co-expression analysis to reveal a previously missing protease for the maturation of class I lanthipeptide paenilan. We further discover widely distributed bacterial M16B metallopeptidases of previously unclear biological function as a new family of lanthipeptide proteases. We show the involvement of a pair of bifunctional M16B proteases in the production of previously unreported class III lanthipeptides with high substrate specificity. Together, these results demonstrate the strength of our correlational networking approach to the discovery of hidden lanthipeptide proteases and potentially other missing enzymes for natural products biosynthesis.

[1] Department of Chemistry and Biochemistry, University of South Carolina, Columbia, SC, USA. [2] Department of Chemistry and The Swire Institute of Marine Science, The University of Hong Kong, Pokfulam Road, Hong Kong, China. [3] Southern Marine Science and Engineering Guangdong Laboratory (Guangzhou), Guangzhou, China. [4] The Mass Spectrometry Center, Department of Chemistry and Biochemistry, University of South Carolina, Columbia, SC, USA. [5] State Key Laboratory of Applied Organic Chemistry, College of Chemistry and Chemical Engineering, Lanzhou University, Lanzhou, China. [6] Institute of Chemical Biology, Shenzhen Bay Laboratory, Shenzhen, China. [7] Department of Biological Sciences, University of South Carolina, Columbia, SC, USA. [8] Department of Pathology, Microbiology and Immunology, School of Medicine, University of South Carolina, Columbia, SC, USA. [9] These authors contributed equally: Dan Xue, Ethan A. Older, Zheng Zhong. ✉email: yxpli@hku.hk; li439@mailbox.sc.edu

Many natural product biosynthetic gene clusters (BGCs) are found not to harbor a full set of necessary genes and must utilize enzymes encoded elsewhere in the genome for biosynthesis. These hidden enzymes are particularly important when the missing biosynthetic steps are responsible for the installation of critical moieties. For example, the antibiotic gentamicin C complex and the antitumor reagent geldanamycin both rely on remote genes located apart from their BGCs for methylation steps that are essential for their biological activities[1,2]. Additionally, the antibiotic prodigiosin undergoes an oxidative cyclization performed by an enzyme encoded outside of its BGC to produce the more potent cycloprodigiosin[3] and the antibiotic activity of mature microcin C relies on a final proteolytic step by an evolutionarily conserved protease system located out of the BGC as well[4].

Besides individual cases, lanthipeptides as an entire class of natural products also frequently rely on unclustered biosynthetic genes for maturation. As one of the most common ribosomally synthesized and post-translationally modified peptides (RiPPs)[5,6], lanthipeptides have been shown to exhibit anti-fungal[7], anti-HIV[8], and antinociceptive activities[9], as well as broad antimicrobial activity against multi-drug-resistant (MDR) bacteria[10,11]. The biosynthesis of all five different classes of lanthipeptides involves a crucial protease-mediated cleavage between the leader and core peptides for final maturation. However, only two types of proteases have been relatively well studied. The first is the subtilisin-like serine protease LanP, employed by class I and II lanthipeptides[12–14]. The second is the papain-like cysteine protease domain of the LanT transporter protein, involved exclusively in the biosynthesis of class II lanthipeptides[14–17]. Due to the absence of protease-encoding genes in most characterized class III and IV lanthipeptide BGCs, the maturation of these two classes is barely understood[18], with FlaP[19] and AplP[20] being recently reported as potential proteases for class III lanthipeptides. With the recent explosion of sequenced microbial genomes, increasingly more lanthipeptide BGCs are being identified as lacking BGC-associated genes to encode proteases[18,21,22]. A missing link between these BGCs and their hidden proteases hinders the discovery, heterologous production, and bioengineering of these potentially bioactive lanthipeptides.

In this study, we hypothesize that lanthipeptide BGCs without any colocalized protease-encoding genes may rely on proteases encoded elsewhere in the genome. Here, we develop a genome mining workflow and use correlation analysis complemented by co-expression analysis to establish the first global correlation network between lanthipeptide precursor peptides and proteases from 161,954 bacterial genome sequences. This correlation network provides guidance for the targeted discovery of hidden lanthipeptide proteases encoded by genes outside of the BGCs. As a proof of principle, we select two representative correlations from the network for study, leading to a simultaneous discovery of previously unreported lanthipeptides and responsible hidden proteases. Particularly, a family of bacterial M16B metallopeptidases with previously unclear biological functions is identified as being responsible for the maturation of several previously unreported class III lanthipeptides.

## Results

### Establishment of a global correlation network between lanthipeptide precursor peptides and proteases. We established a global precursor-protease network for all class I–IV lanthipeptides using 161,954 bacterial genomes obtained from the NCBI RefSeq database. When we initiated our analysis, we had not noticed any reported class V lanthipeptides, thus, this new class was not included for analysis of this report. Analyzing a large

number of genomes with antiSMASH 5.0[23], we identified 21,225 putative lanthipeptide BGCs widely distributed across all bacterial taxa. These BGCs harbor 29,489 highly diverse precursors (Supplementary Data 1). We analyzed genomic regions 10 kb up- and downstream of LanC-like proteins and observed that approximately one-third of these genomic regions do not harbor any protease genes, especially class III system (Supplementary Fig. 1). BGCs without any colocalized protease-encoding genes, may rely on proteases encoded outside of those BGCs for leader peptide removal[5,16,24]. In this scenario, we hypothesized that the specificity between precursors and corresponding proteases still exists, at least to some extent, based on two observations: (i) only certain homologs of a protease in a genome have proteolytic activity against a specific precursor[25,26], and (ii) the proteolytic activity is affected by core peptide modifications[19].

Thus, based on the specificity between precursors and proteases, we performed a global Spearman's rank-order correlation analysis to associate precursors with lanthipeptide proteases (regardless of being encoded inside or outside of a BGC), with an emphasis on the identification of hidden lanthipeptide proteases. Spearmen's correlation accounts for the ranking of data instead of exact values and does not require a normal distribution of the data, which matches the gene distribution in our data. Due to the large number of proteases contained in the genomes performing general functions, it was not practical to directly correlate lanthipeptide precursors to all the proteases in the genomes. Thus, we started with pathway-specific proteases encoded by lanthipeptide BGCs. We hypothesized that pathway-specific proteases likely evolved from general proteases encoded elsewhere in the genome and, at the large scale of the dataset, pathway-specific proteases could collectively represent most functional domains of hidden proteases encoded outside of the BGCs. Indeed, by searching proteases from 21,225 putative lanthipeptide BGCs, we generated a library of 44,260 prospective lanthipeptide proteases, representing 120 unique Pfam[27] domains (Supplementary Table 1). In contrast, previously characterized proteases associated with lanthipeptides only encompass 6 Pfam domains (Supplementary Table 1). We used these 120 Pfam domains to search for proteases from the full set of 161,954 bacterial genomes, resulting in 23,777,967 putative lanthipeptide-related proteases. Grouping these proteases based on their sequence similarity using MMseqs2[28] led to 288,416 groups of proteases. Among these, 10,263 groups each containing 100 or more members were selected for downstream analysis. We also applied the same clustering approach to 29,489 lanthipeptide precursors, forming 4527 groups. Among these, 263 groups each containing ten or more precursors were selected for downstream analysis (Fig. 1a). We then sought to identify links between the selected groups of proteases and precursors. To reduce the effect of phylogenetic relatedness, we performed the correlation analysis individually for each genus, leading to the identification of 5209 significant correlations ($\rho > 0.3$, pAdj < 1E−5, Fig. 1b) between precursors and proteases. These significant correlations ranged from 6 phyla and 114 genera, suggesting that widely distributed proteases, even encoded by genes outside of the BGCs, may function against specific lanthipeptide precursors.

We next focused on class III lanthipeptides due to their elusive maturation process[18]. We identified 1833 putative precursors encoding class III lanthipeptides. These precursor peptides were classified into 217 groups based on sequence similarity, including 18 groups that each contained at least 10 members. We selected the correlations identified in at least ten genomes ($I \geq 10$) for further analysis, leading to prioritization of 91 significant correlations ($\rho > 0.3$, pAdj < 1E−5, $I \geq 10$) between 8 groups of precursors (Supplementary Fig. 2) and 87 groups of proteases. These significant correlations were distributed in two phyla,

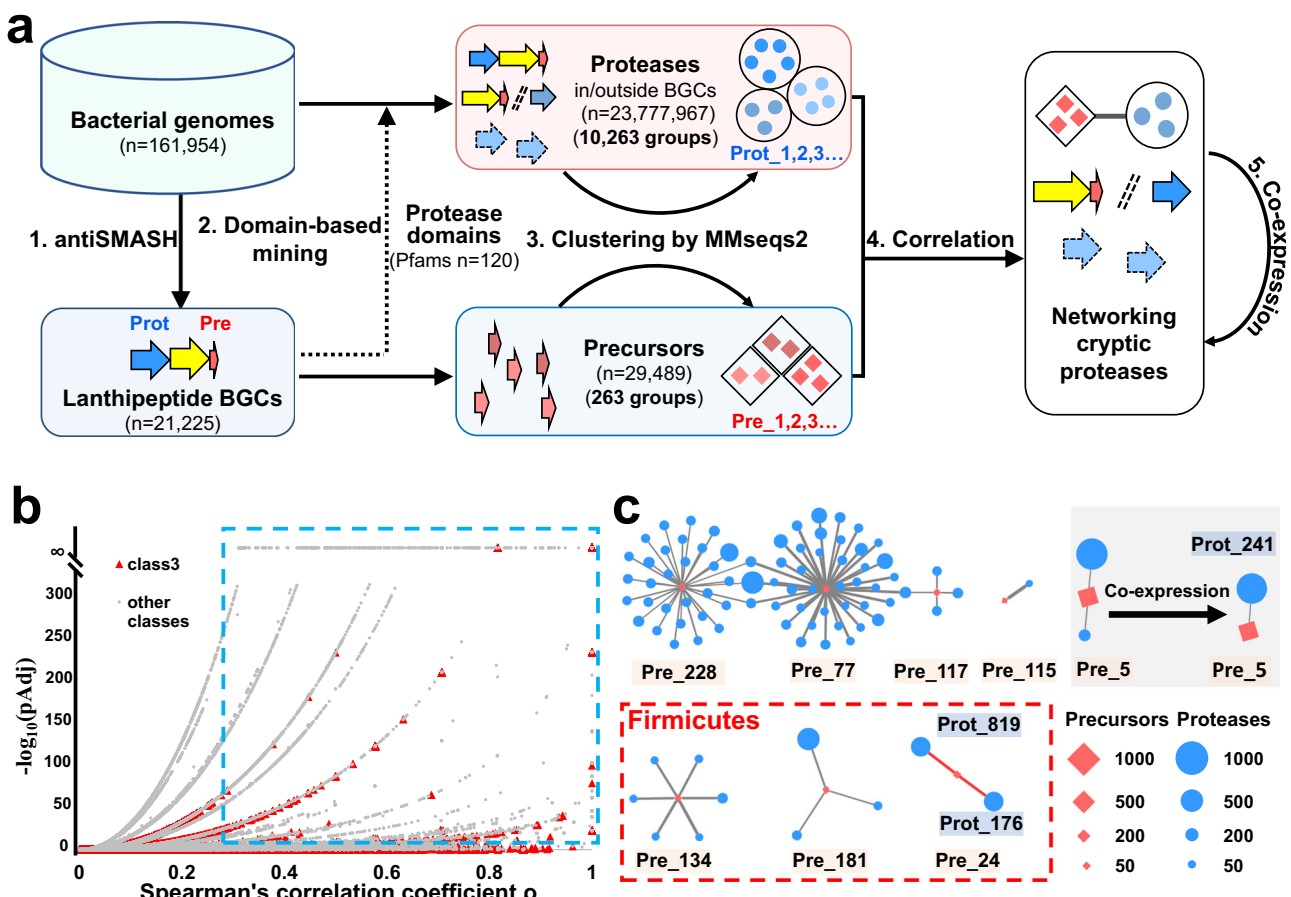

**Fig. 1 Overview of the precursor-protease correlational networking for the discovery of hidden lanthipeptide proteases. a** A genome-mining workflow for discovering potential hidden lanthipeptide proteases, including four steps: (1) antiSMASH 5.0 was used to analyze 161,954 bacterial genomes for identification of lanthipeptide BGCs and BGC-associated proteases; (2) a Pfam domain-based mining of proteases (either inside or outside of BGCs) was performed against entire bacterial genomes based on the 120 Pfams of BGC-associated proteases identified in step (1); (3) Proteases and precursors were clustered by sequence similarity using MMseqs2; (4) Correlation networks at the genomic level were constructed to link precursors and hidden proteases that are not encoded by the BGCs; and (5) Co-expression analysis was integrated to refine the genomic-level correlation of interest. Pre: precursor peptides; Prot: proteases. **b** Volcano plot of 81,323 correlations at the genus level. The blue box encompasses 5209 significant correlations. Red triangles indicate correlations for class III lanthipeptides. *P*-values are calculated by one-sided *t*-test and adjusted by false-discovery rate. **c** Prioritized significant correlations ($\rho > 0.3$, pAdj < 1E−5, one-sided *t*-test, adjusted by false-discovery rate) for class III lanthipeptides from eight genera. Only correlations identified in at least ten genomes ($I \geq 10$) were prioritized. Red diamonds represent groups of precursors and blue circles represent groups of proteases. Shape sizes are proportional to the number of precursors or proteases identified in a group at the genus level. Edges connecting diamonds and circles indicate a significant correlation, with the increasing thickness of the edge representing the increasing strength of the correlation (Spearman's $\rho$). The 11 correlations in Firmicutes (indicated in red box) were formed between three groups of class III precursors (Pre_24, 134, and 181, totally 68 precursors) and 11 groups of peptidases. The remaining 80 correlations found in Actinobacteria were formed between five groups of class III precursors (Pre_5, 77, 115, 117, and 228, totally 796 precursors) and 76 groups of proteases. Specifically, Pre_5, the most abundant precursor group among the prioritized significant correlations, was strongly correlated with two groups of proteases, Prot_241 and Prot_1365. An integration of co-expression analysis reduced the two correlations to only one group, Prot_241.

including 80 correlations in Actinobacteria and 11 correlations in Firmicutes. The core information of these prioritized significant correlations is summarized in Fig. 1c and Supplementary Table 2, with some representative discoveries described below.

Among the 91 significant correlations, metallopeptidases appeared in many significant correlations, suggesting that the metallopeptidase superfamily may play an important role in the maturation of class III lanthipeptides. In contrast, families of serine protease and cysteine protease have been reported for the maturation of class I and II lanthipeptides[14,17,24,29]. Remarkably, among the 91 significant correlations representing 864 precursors, 758 precursors (88%) were strongly correlated with only one or two groups of proteases. For example, the precursors of groups Pre_5, Pre_24, and Pre_115 were only correlated to two,

two, and one group(s) of proteases ($\rho > 0.3$, pAdj < 1E−5, $I \geq 10$; Fig. 1c and Supplementary Table 2) at the genus level of *Streptomyces*, *Paenibacillus*, and *Lentzea*, respectively. None of the 888 proteases correlated to Pre_5, Pre_24, or Pre_115 had been characterized and only 19 of them are encoded by genes within the corresponding BGCs. Thus, this result exhibited the potential of our correlation network in identifying hidden proteases for the maturation of class III lanthipeptides.

On the other hand, five groups of precursors, Pre_77, Pre_117, Pre_134, Pre_181, and Pre_228, were each correlated to multiple groups of proteases, forming five multiple-correlation clusters at the genus level of *Amycolatopsis*, *Streptomyces*, *Alkalihalobacillus*, *Lactobacillus*, and *Rhodococcus*, respectively (Fig. 1c). Together, these five groups only represent the remaining 12% of the

precursors from the 91 significant correlations. Some groups of proteases, e.g., Prot_1169, Prot_2308, and Prot_9513, were observed to share similar functional domains, which may partially account for their simultaneous correlations with the same precursor group. At first glance, this multiple-correlation pattern presented a challenge to identify a responsible protease. However, the correlation strength between different groups of proteases and the same group of precursors appeared to be different based on Spearman's rank correlation coefficient ($\rho$). In addition, we noticed that multiple-correlation clusters could be distinguished by the classes of their correlated proteases. For example, proteases associated with Pre_117 belong primarily to the α/ß hydrolase class while those associated with Pre_134 mainly fall into the metallopeptidase class (Supplementary Table 2). For the multiple-correlation pattern, to complement correlational analysis at the genomic level, we also integrated co-expression analysis using whole genome transcriptomic data to prioritize target proteases at the intersection of correlational and co-expression analyses. For example, regarding the aforementioned Pre_5 that was strongly correlated with two groups of proteases (Fig. 1c), Prot_241 (metal-dependent hydrolase; PF10118) and Prot_1365 (carbon-nitrogen hydrolase; PF00795), we compiled publicly available transcriptomic data consisting of 80 samples from three *Streptomyces* strains that expressed Pre_5. Co-expression analysis using these data reduced the original two strong correlations to only one group, Prot_241 ($\rho > 0.4$, pAdj < 0.05). Thus, we demonstrate here the establishment of a global correlation network, integrating co-expression data to fortify lanthipeptide precursor-protease association predictions.

**Validation of the precursor-protease correlation network**. We used previously characterized representative lanthipeptide proteases to validate the correlation network by first examining better-understood class I and II lanthipeptide systems. In our network, we located the previously studied class I lanthipeptides epidermin[30] and gallidermin[30] in one group, Pre_1, due to their precursor similarity, while class II lanthipeptides thuricin[31], cerecidin[32], and cytolysin[33] were found in four groups of precursors, Pre_11, Pre_12, Pre_10, and Pre_13 (cytolysin is comprised of two distinct precursors resulting in its representation in two precursor groups) (Supplementary Fig. 3). The BGCs producing these lanthipeptides also encode pathway-specific LanP and LanT homologs, two types of proteases known to be involved in class I and II lanthipeptide biosynthesis[14,16,34] (Supplementary Fig. 4). Indeed, these LanP and LanT homologs each were shown in our network to correlate with Pre_1, Pre_11, Pre_12, Pre_10, and Pre_13, respectively (Supplementary Table 3). Additionally, we used the previously characterized class III lanthipeptide protease FlaP for validation. Besides the genera of *Kribbella* and *Streptomyces* that were reported to harbor a FlaP-like protease colocalized with a class III precursor[19], we identified FlaA homologs in additional genera, mainly *Amycolatopsis* and *Jiangella* (Supplementary Fig. 5). FlaP-like proteases were simultaneously identified in our network to significantly correlate with FlaA in *Amycolatopsis* and *Jiangella*, with $\rho = 0.77$, pAdj = 3E−16 and $\rho = 1$, $p = 8E−78$, respectively (Supplementary Data 1). Such strong correlations supported the effectiveness of applying our correlation network to class III lanthipeptides.

**Identification of a hidden protease for maturation of class I lanthipeptide paenilan**. To identify lanthipeptide proteases that are not encoded by their respective BGCs, we first selected a group of class I lanthipeptide precursors, Pre_49 (Fig. 2a), which presumably encodes paenilan as a representative product based on the precursor sequences[35] (Fig. 2b). All 44 BGCs of this group

do not harbor any protease-encoding genes within the BGCs and the responsible proteases for maturation of this group of lanthipeptides have not been reported[35], consistent with the recent finding that increasingly more class I lanthipeptide BGCs have been noticed to lack a pathway-specific protease[16,21,22]. In our network, 34 groups of proteases were shown to be strongly correlated with Pre_49 ($\rho > 0.3$, pAdj < 1E−5, $I \geq 10$). Additionally, integrating transcriptomic data of 18 samples from *Paenibacillus polymyxa* ATCC 842 expressing the precursor of Pre_49 (Supplementary Data 1), we looked for the intersection of proteases identified by both correlational and co-expression analyses and reduced the putative correlations from 34 groups to four groups, Prot_686, Prot_1570, Prot_2941, and Prot_8704 (Fig. 2c). Thus, we expressed the corresponding proteases of these four groups from *P. polymyxa* ATCC 842 as recombinant proteins and tested their proteolytic activity in vitro against the PllB/PllC-modified precursor peptide from the same strain. LC-MS analysis revealed that paenilan, the same product as the native strain, was produced in the assay with the member of Prot_686 (Fig. 2d). Prot_686 belongs to the S8 family of peptidases. This family has been previously reported for the maturation of class I lanthipeptides[12,24]. Thus, we used the correlation of Prot_686 with a paenilan-producing precursor to demonstrate that proteases predicted by our network analysis could in fact be responsible for lanthipeptide maturation. This motivated us to further exploit the power of our network in the less understood class III lanthipeptide system.

**Networking analysis reveals a new family of lanthipeptide proteases and links it to the production of previously unreported class III lanthipeptides**. Class III lanthipeptide BGCs are abundant in Firmicutes[6]. However, the lack of a protease-encoding gene in most class III BGCs has been a challenge toward exploiting these lanthipeptides[18], such as heterologous expression for the discovery of new class III lanthipeptides, pathway bioengineering for increasing production yield, chemical diversity, and biological activities, and leveraging the enzymology of proteases as a synthetic biology tool for general proteolytic and traceless tag removal applications[14]. Therefore, we focused on Firmicutes for a simultaneous discovery of previously unreported class III lanthipeptides and their associated hidden proteases. Our attention was drawn to two groups of proteases designated as Prot_176 and Prot_819, both of which had not been studied as lanthipeptide proteases and showed strong correlations ($\rho = 0.69$, pAdj = 4E−64) to a group of uncharacterized precursors, Pre_24 (Fig. 1c). Pre_24, consisting of total 115 precursors, is solely distributed in Firmicutes and represents the largest precursor group in Firmicutes (Fig. 3a). These 115 precursors are distributed across 48 class III lanthipeptide BGCs from three genera, *Alkalihalobacillus*, *Bacillus*, and *Paenibacillus*. The strains harboring *pre_24* all possess one pair of *prot_176* and *prot_819* present side by side outside of the 48 BGCs (Fig. 3b). Among these 48 BGCs, an extra pair of *prot_176* and *prot_819* was identified within 17 BGCs (Fig. 3b). In addition, many other strains from *Alkalihalobacillus*, *Bacillus*, and *Paenibacillus* that do not encode Pre_24 also contain a pair of *prot_176* and *prot_819* in their genomes (Fig. 3b).

To investigate Prot_176 and Prot_819 as potential proteases for Pre_24, we first produced and characterized representative class III lanthipeptides from Pre_24. From the network, we selected two Pre_24-encoding BGCs for study. The first was from *Bacillus nakamurai* NRRL B-41092, designated as *bcn*. The gene organization of the *bcn* BGC is shown in Fig. 3c, with *bcnA1* and *bcnA2* corresponding to *pre_24*. The *bcn* BGC does not harbor any protease-encoding genes. Instead, outside of the *bcn*

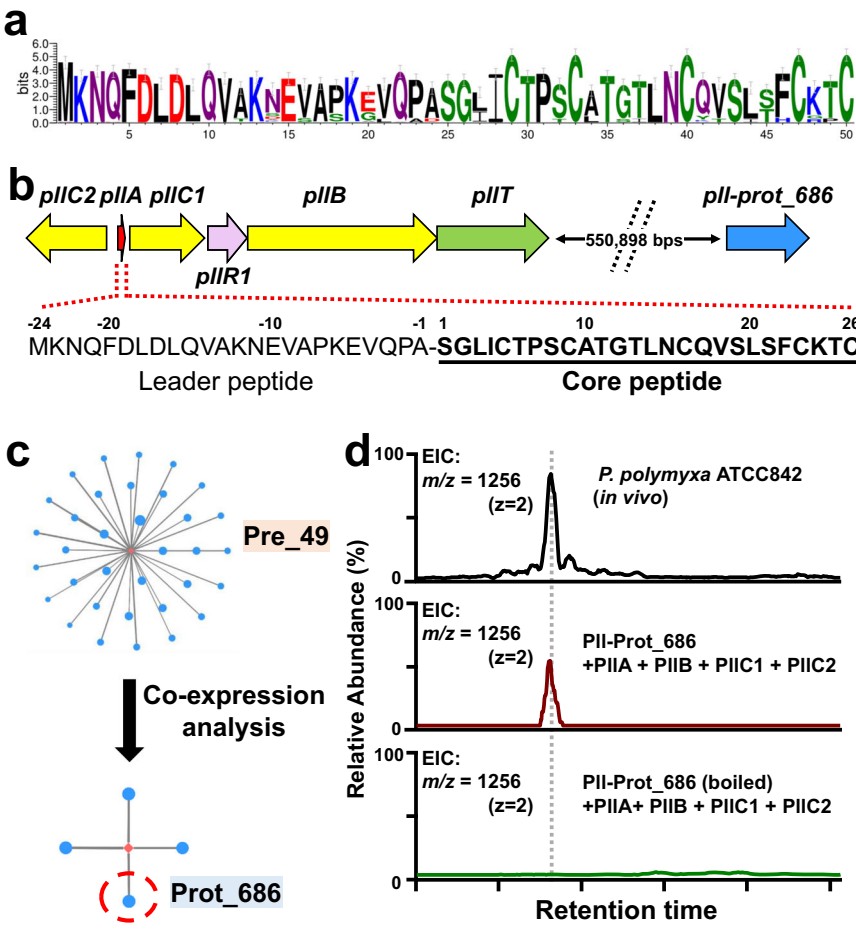

**Fig. 2 Identification of a hidden protease from Prot_686 for maturation of paenilan. a** Sequence logo of the group Pre_49. Sequence alignments were trimmed by trimAl to remove gap-rich regions before drawing logo. Error bars indicate sample correction, and the total height of the error bar is twice this correction. **b** A representative BGC of Pre_49 in *P. polymyxa* ATCC 842 that presumably producing paenilan based on the precursor sequence. All 44 BGCs of this group do not harbor any protease-encoding genes, suggesting hidden proteases encoded elsewhere in the genome. Dashed lines separating *pllT* and *prot_686* indicate a gap in their respective genetic locations. This type of notation is also used in other figures below. The distance of this gap, 550,898 base pairs, is represented here to convey magnitude. **c** Intersection of precursor-protease correlation (*ρ* > 0.3, pAdj < 1E−5, one-sided *t*-test, adjusted by false-discovery rate, *l* ≥ 10) and co-expression (*ρ* > 0.4, pAdj < 0.05, one-sided *t*-test, adjusted by false-discovery rate) suggested candidate proteases(s). **d** LC-MS analysis indicated the proteolytic activity of Prot_686 for the maturation of paenilan. EIC: extracted ion chromatogram.

BGC, a pair of protease-encoding genes was identified as belonging to *prot_819* and *prot_176*, respectively (Figs. 1c and 3c). This pair of genes was designated as *bcn-genomeP1* (abbreviation *bcn-gP1* hereafter) and *bcn-genomeP2* (abbreviation *bcn-gP2* hereafter). The *bcn* BGC, together with *bcn-gP1* and *bcn-gP2*, was constructed into a pDR111-based integrative plasmid (Supplementary Table 4) and heterologously expressed in the host *Bacillus subtilis* 168, leading to the detection of two lanthipeptides named bacinapeptins A and B (Supplementary Figs. 6 and 7). These compounds were also detected in the native strain. Due to limited amounts of bacinapeptins A and B produced, we used high-resolution mass spectrometry (HRMS) and tandem MS (MS/MS) (Supplementary Figs. 6 and 7), in combination with the sequences of BcnA1 and BcnA2, to establish the chemical structures of bacinapeptins A (presumably derived from BcnA1) and B (presumably derived from BcnA2), as shown in Fig. 3d. The structure elucidation is described in the Supplementary Note.

The second BGC selected was from *Paenibacillus thiaminolyticus* NRRL B-4156, designated as *ptt*. The gene organization of the *ptt* BGC is shown in Fig. 3e, with *pttA1-A7* belonging to *pre_24*. Outside of the *ptt* BGC, there is a pair of protease-encoding genes, *ptt-genomeP1* (hereafter abbreviated as *ptt-gP1*)

and *ptt-genomeP2* (hereafter abbreviated as *ptt-gP2*), that belongs to *prot_819* and *prot_176*, respectively (Figs. 1c and 3e). Within the *ptt* BGC, there is an additional pair of protease-encoding genes, *pttP1* and *pttP2*, also belonging to *prot_819* and *prot_176*, respectively. The *ptt* BGC, with *pttP1* and *pttP2*, or with *ptt-gP1* and *ptt-gP1* replacing *pttP1* and *pttP2*, was cloned into pDR111 (Supplementary Table 4) and heterologously expressed in the host *B. subtilis* 168, respectively, both leading to the detection of five lanthipeptides named paenithopeptins A–E (Fig. 3f and Supplementary Fig. 8), with the former one showing higher production yield. These compounds were also produced by the native strain. We isolated paenithopeptin A, the most abundant product, and established its chemical structure via HRMS, MS/MS fragmentation, Marfey's analysis, and 1D and 2D NMR (Supplementary Figs. 9–20 and Supplementary Table 6). Detailed structure elucidation is described in the Supplementary Note. Paenithopeptins B–E identified by MS/MS fragmentation analysis (Supplementary Figs. 21–24) have different overhangs of amino acids at the N-terminus compared to paenithopeptin A. The chemical structures of paenithopeptins A–E suggest that they were all derived from the precursor PttA1. Notably, paenithopeptins A–E and bacinapeptins A and B represent previously

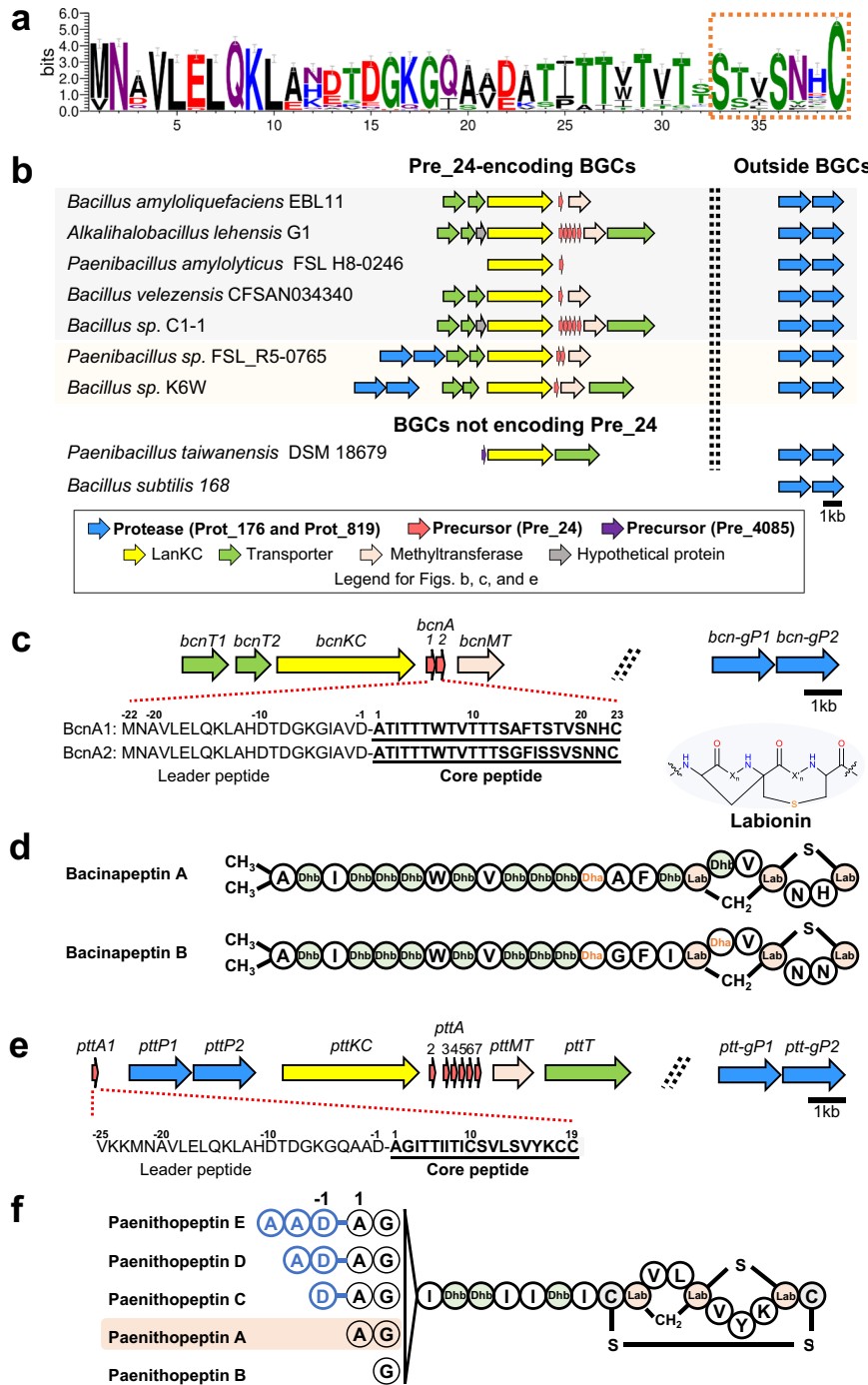

**Fig. 3 Discovery of a family of previously unknown lanthipeptide proteases linked to the production of previously unreported class III lanthipeptides. a** Sequence logo of Pre_24. Sequence alignments were trimmed by trimAl to remove gap-rich regions before drawing logo. The putative core peptides contain only one S-(X)$_2$-S-(X)$_{2-5}$-C motif (X stands for random amino acid) as opposed to two or three such motifs commonly seen in previously known class III lanthipeptides. Error bars indicate sample correction, and the total height of the error bar is twice this correction. **b** Representative class III lanthipeptide BGCs encoding Pre_24, showing distribution of a pair of protease-encoding genes, *prot_176* and *prot_819*, outside of the BGCs and an extra pair in a minority of the BGCs. In addition, a pair of *prot_176* and *prot_819* is also present in the genomes of many strains that do not harbor Pre_24-encoding BGCs. **c** A Pre_24-encoding BGC, *bcn*, selected for study. The *bcn* BGC putatively encodes a class III lanthipeptide synthetase (BcnKC), two precursors (BcnA1 and BcnA2), a methyltransferase (BcnMT), and a transporter (BcnT). A pair of protease-encoding genes belonging to *prot_819* and *prot_176*, respectively, is absent in the *bcn* BGC, but present elsewhere in the genome, designated as *bcn-genomeP1* (abbreviation *bcn-gP1*) and *bcn-genomeP2* (abbreviation *bcn-gP2*). **d** Structures of bacinapeptins A and B produced by the *bcn* BGC in the presence of *bcn-gP1* and *bcn-gP2*. **e** Another Pre_24-encoding BGC, *ptt*, selected for study. The *ptt* BGC putatively encodes a characteristic class III lanthipeptide synthetase (PttKC) and seven precursors, PttA1-PttA7. While *pttA2-pttA7* are together, *pttA1* is located separately. The putative protease-encoding genes *pttP1* and *pttP2* belonging to *prot_819* and *prot_176*, respectively, are adjacent to each other downstream of *pttA1*. An extra pair of protease-encoding genes also belonging to *prot_819* and *prot_176* is located outside of the *ptt* BGC, designated as *ptt-genomeP1* (abbreviation *ptt-gP1*) and *ptt-genomeP2* (abbreviation *ptt-gP2*), respectively. *pttMT* and *pttT* are predicted to encode a methyltransferase and a transporter, respectively. **f** Structures of paenithopeptins A–E produced by the *ptt* BGC.

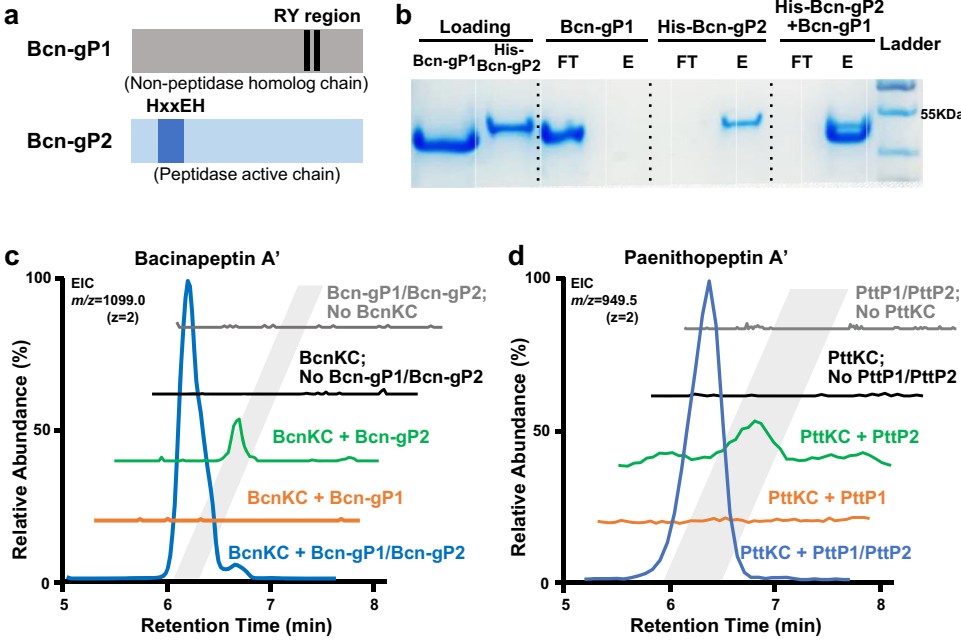

**Fig. 4 Bioinformatics analysis and enzymatic characterization of Bcn-gP1/Bcn-gP2 and PttP1/PttP2. a** Representation of the $Zn^{2+}$-binding HXXEH motif of Bcn-gP2 and PttP2 as well as the substrate-binding R/Y region of Bcn-gP1 and PttP1. These characteristic features suggested Bcn-gP1/Bcn-gP2 and PttP1/PttP2 as heteromeric M16B metallopeptidases. **b** A pull-down assay showing protein-protein interaction between Bcn-gP1 and Bcn-gP2. $His_8$-tag-free Bcn-gP1 was immobilized on the nickel affinity column only when $His_8$-tagged Bcn-gP2 was present, as reflected by examining the flowthrough using sodium dodecyl sulfate polyacrylamide gel electrophoresis (SDS-PAGE). The experiment was repeated three times independently with similar results. FT: flowthrough; E: elute. **c** Extracted ion chromatogram (EIC) showing in vitro characterization of the proteolytic activity of Bcn-gP1/Bcn-gP2. Bcn-gP1 and Bcn-gP2 were required simultaneously to produce the highest yield of bacinapeptin A′. In addition, the lack of BcnKC in the assay completely abolished the production, suggesting that Bcn-gP1/Bcn-gP2 shows specificity towards BcnKC-modified precursor bearing a labionin ring. In the assay without BcnKC, $m/z$ 1216.1 ($z = 2$) was also used for EIC detection of potential unmodified core peptide. **d** Extracted ion chromatogram (EIC) showing in vitro characterization of the proteolytic activity of PttP1/PttP2. PttP1 and PttP2 were required simultaneously for the highest production yield and the lack of PttKC abolished the production. In the assay without PttKC, $m/z$ 994.5 ($z = 2$) was also used for EIC detection of potential unmodified core peptide.

unreported class III lanthipeptides. The tricyclic ring system (labionin in a larger disulfide-bridged ring) in paenithopeptins A–E has not been previously reported in Firmicutes.

Next, we used the pair of Bcn-gP1 and Bcn-gP2 and the pair of PttP1 and PttP2 as examples to confirm the proteolytic activity of Prot_819 and Prot_176 against Pre_24 as the precursor peptide. We selected these two pairs of proteases because: (i) they both belong to Prot_819 and Prot_176, and (ii) while Bcn-gP1 and Bcn-gP2 are encoded by protease genes outside of the *bcn* BGC, PttP1 and PttP2 are encoded by protease genes within another BGC *ptt*, representing both scenarios of protease gene distribution, outside or inside of BGCs. We began the investigation with bioinformatics analysis of Bcn-gP1 and Bcn-gP2, which suggested that both belong to the family of zinc-dependent M16 peptidases. Specifically, Bcn-gP2 possesses an HXXEH motif essential for $Zn^{2+}$ binding and catalytic activity, while Bcn-gP1 contains an R/Y pair in the C-terminal domain to facilitate substrate binding (Fig. 4a). The characteristic sequences of Bcn-gP1 and Bcn-gP2 are highly reminiscent of two known M16B peptidases from *Sphingomonas* sp. A1, Sph2681 and Sph2682, that form a heterodimer[36,37]. To investigate this, we performed homology modeling of Bcn-gP1 and Bcn-gP2, using the heterodimeric crystal structure of Sph2681/Sph2682 (PDB code 3amj) as the template. The best model of the Bcn-gP1 and Bcn-gP2 heterodimer (hereafter called Bcn-gP1/Bcn-gP2) displayed a high degree of structural similarity toward the template structure (Supplementary Fig. 25). Likewise, bioinformatics analysis of PttP1 and PttP2 showed consistent results (Supplementary Fig. 26). We further performed pull-down assays and confirmed the respective protein–protein interaction between Bcn-gP1 and

Bcn-gP2 (Fig. 4b) as well as PttP1 and PttP2 (Supplementary Fig. 27). Taken together, these results suggested that Bcn-gP1/Bcn-gP2 and PttP1/PttP2 each functions as a heteromeric M16B metallopeptidase.

We then performed an in vitro characterization of Bcn-gP1/Bcn-gP2 against BcnA1. We individually expressed and purified recombinant His-tagged BcnA1, BcnKC, Bcn-gP1, or Bcn-gP2. We then incubated BcnA1, with or without BcnKC, followed by adding Bcn-gP1, Bcn-gP2, or both, respectively, for an incubation of 12 h. We observed that (i) while excluding Bcn-gP1 generated a detectable production presumably due to Bcn-gP2 possessing catalytic residues, Bcn-gP1 and Bcn-gP2 were simultaneously required to generate the highest yield of the product, demethylated bacinapeptin A (hereafter called bacinapeptin A′, produced due to not including the predicted methyltransferase BcnMT in the assay); and (ii) the Bcn-gP1/Bcn-gP2 complex has specificity against BcnKC-modified precursor bearing a labionin ring (Fig. 4c). Likewise, we performed an in vitro characterization of PttP1/PttP2 against PttA1, with or without PttKC, using the same assay conditions mentioned above. We observed consistent results (Fig. 4d) as described for Bcn-gP1/Bcn-gP2. Notably, the products of this in vitro assay, designated as paenithopeptins A′-E′, each lacked a disulfide bond compared to paenithopeptins A–E, presumably due to the absence of a disulfide bond-forming oxidoreductase[38,39] that is contained in the paenithopeptin-producing bacteria. We further performed an in vivo assay of *pttP1*/*pttP2*, aiming to produce intact paenithopeptins A–E to confirm the function of *pttP1*/*pttP2*. Different combinations of four genes, *pttA1*, *pttKC*, *pttP1*, and *pttP2*, were constructed into a series of pDR111-based integrative plasmids (Supplementary

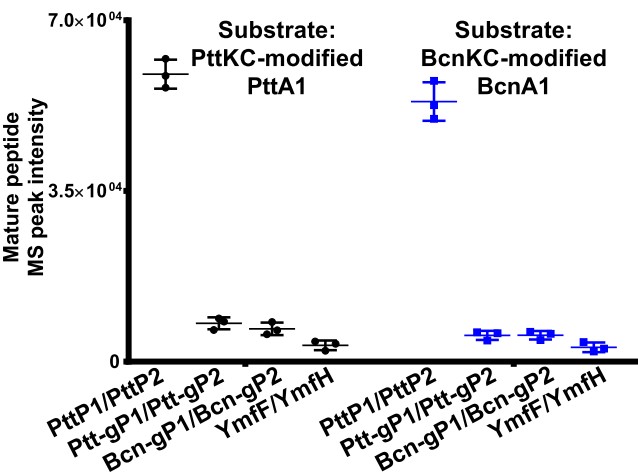

**Fig. 5 Different efficiencies of Prot_819/Prot_176 members.** PttP1/PttP2, Ptt-gP1/Ptt-gP2, Bcn-gP1/Bcn-gP2, and YmfF/YmfH against PttKC-modified PttA1 and BcnKC-modified BcnA1, respectively, with PttP1/PttP2 showing the highest efficiency in both cases. Error bars indicate standard deviation across triplicates. Data shown as mean ± SD (each group $n = 3$; SD: standard deviation).

Table 4) and expressed in an engineered host *B. subtilis* 168 with *ymfF* and *ymfH* (homologs of *pttP1* and *pttP2*, respectively) deleted in advance. The in vivo assay indeed showed the production of fully modified paenithopeptins (Supplementary Fig. 27), in a production trend consistent with the in vitro assay. Taken together, these results indicated Prot_819/Prot_176 as previously unknown lanthipeptide proteases and established their role as a heteromeric complex in the maturation of previously unreported class III lanthipeptides represented by bacinapeptins and paenithopeptins.

**Different efficiencies of Prot_819/Prot_176 proteases suggest potentially evolved protease specificity for class III lanthipeptides.** Due to the wide distribution of *prot_819/prot_176* members in bacterial genomes, we investigated the potential activity of different Prot_819/Prot_176 members for leader removal. The distribution of *prot_819/prot_176* pairs were classified into three scenarios for our comparison (Fig. 3b): (i) outside of Pre_24-encoding class III lanthipeptide BGCs, e.g., *bcn-gP1/bcn-gP2* from *B. nakamurai* NRRL B-41092 and *ptt-genomeP1/ptt-genomeP2* (abbreviated as *ptt-gP1/ptt-gP2*) from *P. thiaminolyticus* NRRL B-4156, (ii) within the Pre_24-encoding class III lanthipeptide BGCs, e.g., *pttP1/pttP2* within the *ptt* BGC, and (iii) in the genomes that do not harbor any Pre_24-encoding class III lanthipeptide BGCs, e.g., *ymfF/ymfH* from *B. subtilis* 168. We compared the in vitro proteolytic activity of these Prot_819/Prot_176 members using the same assay conditions described above. The results show that YmfF/YmfH, Bcn-gP1/Bcn-gP2, and Ptt-gP1/Ptt-gP2 were able to cleave PttKC-modified PttA1 and produce paenithopeptins, but the efficiency, reflected as the production yield under the same conditions, was ~4, ~13, and ~15% compared to that of PttP1/PttP2, respectively (Fig. 5 and Supplementary Fig. 28). We further evaluated the activity of these Prot_819/Prot_176 proteases against an additional member of Pre_24, i.e., BcnA1 from *B. nakamurai* NRRL B-41092. As expected, we observed that YmfF/YmfH, Bcn-gP1/Bcn-gP2, Ptt-gP1/Ptt-gP2, and PttP1/PttP2 were all active against BcnKC-modified BcnA1, with an efficiency trend (Fig. 5 and Supplementary Fig. 29) similar to that against PttKC-modified PttA1.

On the other hand, we tested whether the Prot_819/Prot_176 members could act on precursors from groups other than Pre_24.

The precursor selected from a class III lanthipeptide BGC of *Paenibacillus taiwanensis* DSM 18679 was designated as PbtA (belonging to Pre_4085) (Fig. 3b). We expressed PbtA and Pbt-genomeP1/Pbt-genomeP2 (abbreviation Pbt-gP1/Pbt-gP2) as recombinant proteins for in vitro proteolytic assays. Despite a variety of assay conditions used, we did not observe any obvious maturation products when PbtA, with or without PbtKC, was incubated with Pbt-gP1/Pbt-gP2, YmfF/YmfH, Bcn-gP1/Bcn-gP2, Ptt-gP1/Ptt-gP2, or PttP1/PttP2.

Taken together, the specificity between Pre_24 and Prot_819/Prot_176 supports our precursor-protease correlation network. Furthermore, considering the wide distribution of Prot_819/Prot_176, the different efficiencies of Prot_819/Prot_176 members appeared supportive of our hypothesis that certain proteases with general functions in the genome might have evolved more or less specific activity against class III lanthipeptides, and extra copies of them might have further evolved into pathway-specific proteases for enhanced activity and specificity. We, therefore, performed a phylogenetic analysis of Prot_176 members, the catalytic component of the Prot_819/Prot_176 pair. A phylogenetic tree was built at the level of genus *Paenibacillus* where paenithopeptins were isolated. Indeed, pathway-specific Prot_176 appeared in closely related lineages in the phylogenetic tree, forming an independent branch from other Prot_176 members encoded by genes outside of the BGCs, implying a process of gene divergence (Supplementary Fig. 30). Thus, the substrate specificity of Prot_819/Prot_176 likely gained during evolution provides evidence for our correlational networking that uses substrate specificity to look for hidden proteases.

**Member of Prot_819/Prot_176 shows specificity and unique activity for leader peptide removal.** Due to the highest activity of PttP1/PttP2 shown above, we further characterized this protease pair in detail as a representative member of Prot_819/Prot_176 that are widely distributed outside of many class III lanthipeptide BGCs. First, we confirmed the importance of the Zn-binding HXXEH motif and the R/Y pair in the proteolytic activity of PttP1/PttP2 against PttKC-modified PttA1 precursor. We individually mutated H67, E70, and H71 of the HXXEH motif and R298 and Y305 of the R/Y pair to Ala, leading to decreased activity of PttP1/PttP2 in vitro for each mutation (Supplementary Fig. 31). Simultaneous mutation of all these five residues into Ala residues completely abolished the production of paenithopeptins A′–E′ (Supplementary Fig. 31). In addition, the metal-chelating compound, *o*-phenanthroline, significantly inhibited the activity of PttP1/PttP2 (Supplementary Fig. 32).

Next, we investigated how PttP1/PttP2 led to the production of a series of compounds with different N-terminal overhangs from the same precursor peptide, PttA1 (Fig. 6a). We first performed an in vitro enzymatic assay with an incubation of PttA1 and PttKC for four hours, followed by adding both PttP1 and PttP2 for different incubation lengths, ranging from 15 min to 36 h. We observed that at the 15 min-point, paenithipeptin E′ with the longest leader overhang was formed as the major product (Fig. 6b), suggesting an endopeptidase activity of PttP1/PttP2. Over time, paenithipeptin E′ diminished while paenithipeptin A′ (with no leader overhang) was accumulated (Fig. 6b), followed by removal of the N-terminal alanine of paenithipeptin A′ to form paenithipeptin B′ after an extended incubation of 36 h (Fig. 6b). This transformation process suggested an aminopeptidase activity of PttP1/PttP2. Next, we directly used paenithopeptin E′ as a substrate for incubation with PttP1/PttP2, leading to the production of paenithipeptin A′ (Fig. 6c). We also incubated paenithopeptin A with PttP1/PttP2, resulting in the generation of paenithopeptin B (Fig. 6d). These experiments confirmed the

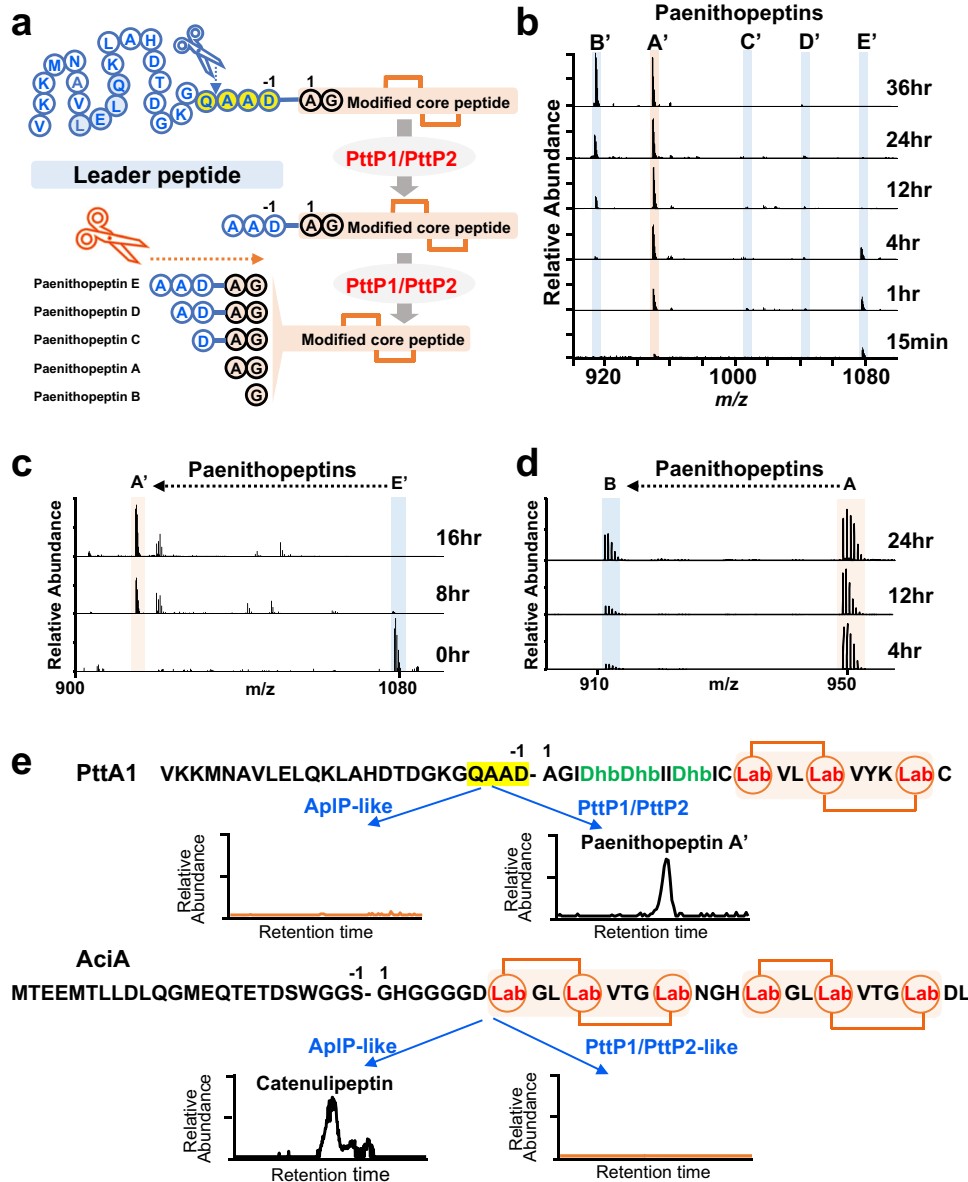

**Fig. 6 PttP1/PttP2 showed substrate specificity and bifunctional proteolytic activity for leader peptide processing. a** Representation of the leader peptide processing by PttP1/PttP2, through an initial endopeptidase activity to partially remove the leader followed by an aminopeptidase activity to successively remove the remaining overhangs. **b** LC-MS analysis of an in vitro assay of PttP1/PttP2, showing production of paenithopeptins A'–E' at different time points of the assay. Each spectrum shows the summed MS over the retention time of 5.75–6.75 min. **c** LC-MS analysis showing the transformation of paenithopeptin E' to paenithopeptin A' by aminopeptidase activity of PttP1/PttP2. Each spectrum shows the summed MS over the retention time of 5.75–6.75 min. **d** LC-MS analysis showing the transformation of paenithopeptin A to paenithopeptin B by aminopeptidase activity of PttP1/PttP2. Each spectrum shows the summed MS over the retention time of 5.75–6.75 min. **e** In vitro assay of PttP1/PttP2-like proteases in comparison with AplP-like proteases recently reported for class III lanthipeptide maturation, indicating that PttP1/PttP2-like and AplP-like proteases have respective substrate specificity. The conserved Q-A-(A/V)-(D/E) motif of Pre-24, absent in the precursor peptides for AplP-like proteases, was specifically cleaved by PttP1/PttP2-like proteases. The spectra shown were EIC at $m/z = 949.5$ ($z = 2$) for paenithopeptin A' and EIC at $m/z = 800.0$ ($z = 3$) for catenulipeptin. The MS spectra in figures **b**–**e** were recorded at positive mode.

aminopeptidase activity of PttP1 and PttP2. Taken together, our results show that the QAAD motif in the leader of PttA1 appeared to be specifically cleaved by PttP1/PttP2, through an initial endopeptidase activity to partially remove the leader followed by an aminopeptidase activity to successively remove the remaining overhangs. We then investigated whether PttP1 and PttP2 have proteolytic activity against PttA2-PttA7. Although *pttA2-pttA7* are located separately from *pttA1* in the *ptt* BGC (Fig. 3c), they also belong to the *pre_24* group in our network. As expected, in vitro enzymatic assays using conditions described

above revealed corresponding products cleaved from the conserved Q-A-(A/V)-(D/E) motif of PttA2-PttA7 (Supplementary Figs. 33–36). The products also showed different overhangs of amino acids at the N-terminus (Supplementary Figs. 37–40), suggesting again both endo- and aminopeptidase activities of PttP1/PttP2. In addition, the formation of these products was also dependent on the precursor modification by PttKC (Supplementary Figs. 33–36).

We further studied substrate specificity of PttP1/PttP2 in comparison with AplP-like proteases recently discovered from

Actinobacteria for the maturation of class III lanthipeptides[20]. Since the native producer of paenithopeptins A–E also harbors an AplP-like protease, we expressed this protease as a recombinant protein and tested whether it could cleave PttA1 (belonging to Pre_24), with or without modification by PttKC. However, although AplP-like proteases were hypothesized as a universal strategy for leader peptide removal of class III lanthipeptides[20], we could not detect any products, suggesting that the most abundant group of class III precursors in Firmicutes was not recognizable by Actinobacteria-derived AplP-like proteases (Fig. 6e). On the other hand, we cloned AciA (an experimentally validated precursor for AplP-like proteases; belonging to Pre_2495) from *Catenulispora acidiphila* DSM 44928 for a comparison experiment. When AciA, with or without modification by AciKC, was incubated with a pair of PttP1/PttP2 homolog from *C. acidiphila* DSM 44928, no products were detected (Fig. 6e). These results indicated that PttP1/PttP2 and AplP-like proteases have respective specificity against different groups of precursors and supported our protease-precursor correlation network.

## Discussion

Using a correlation-guided genome-mining strategy to search for unclustered natural product biosynthetic genes, we established a global network bridging the gap between bacterial lanthipeptide precursors and hidden proteases which are not encoded by their respective BGCs. Our results showcase the promise of precursor-protease correlational networking in targeted discovery of both missing proteases for previously known lanthipeptides (e.g., Prot_686 for the maturation of class I lanthipeptide paenilan) and new families of lanthipeptide proteases involved in the maturation of previously unreported products (e.g., M16B metallopeptidases leading to the production of previously unreported class III lanthipeptides paenithopeptins and bacinapeptins). This proof-of-principle study further suggests the potential of genomic-level correlational networking in discovering unclustered natural product biosynthetic genes.

Many natural product BGCs are found to lack certain key genes and must rely on enzymes encoded outside of the BGCs for biosynthesis. In lanthipeptides, a final protease performs an indispensable maturation step. Notably, with the advancement of sequencing and genome-mining technologies, increasingly more lanthipeptide BGCs lacking protease-encoding genes are being identified, consistent with our data that about one-third of lanthipeptide BGCs do not harbor protease genes. Particularly, most reported class III and IV lanthipeptide BGCs lack protease-encoding genes[18]. Computational workflows have been developed to predict lanthipeptide BGCs with high accuracy by following canonical co-localized biosynthetic gene rules[23,40]. However, these approaches cannot account for hidden proteases located far outside of the BGC. We used functional domain-based genome mining to search for hidden proteases in 161,954 bacterial genomes, expanding the pool of lanthipeptide protease Pfam domains to 120 as opposed to the 6 domains (Supplementary Table 1) of previously reported lanthipeptide proteases LanP[14], LanT[16], FlaP[19], and AplP[20]. Furthermore, using Spearman's rank-order correlation analysis within each bacterial genus, our networking strategy linked hidden proteases to lanthipeptide precursors. As mentioned above, multiple-correlation clusters represent a challenge in determining candidate proteases. Additionally, the 5209 precursor-protease correlations identified in this study may contain weak correlations which would require more data to refine. Addressing this, we integrated co-expression analysis to complement our approach. We envision that with increasing availability of sequenced bacterial genome and

transcriptome data, we can continuously increase the power of our correlation network.

The biological function of widely distributed M16B metallopeptidases has been unclear in bacteria[41]. Our network guided the discovery of a new function of these proteins as lanthipeptide proteases. We characterized representative Prot_819/Prot_176-belonging proteases of the M16B family and showed their high substrate specificity toward lanthipeptide precursors, despite being encoded outside of the respective BGCs. This result supports the basis of the correlational networking approach that hidden proteases exhibit a measurable degree of substrate specificity. Furthermore, varying efficiencies of Prot_819/Prot_176-belonging proteases from different genomic loci supports a process of functional evolution and provides a method to identify higher efficiency proteases for more productive bioengineering of lanthipeptides.

Consistent with the literature[6,42], our study revealed Firmicutes as a prolific source for the discovery of underexplored class III lanthipeptides that often lack pathway-specific proteases. Exploring this feature in the network, we identified previously unreported lanthipeptides paenithopeptins and bacinapeptins which represent the largest precursor group in Firmicutes. Our network further reveals the wide distribution of uncharacterized proteases in Firmicutes as exemplified by Prot_819/Prot_176-belonging proteases. Their discovery and characterization showcase the potential of our networking approach to link hidden proteases to corresponding BGCs.

Taken together, our correlational networking approach takes advantage of readily available data and easily accessible computational platforms which can be developed into a streamlined process to discover hidden proteases for the exploitation of lanthipeptides. Furthermore, with exponentially increasing microbial genome sequences available, our approach is envisioned to be applicable to other natural product classes that lack necessary genes in their respective BGCs.

## Methods

**BGCs, precursors, and proteases detection**. Publicly available bacterial genomes were downloaded from NCBI RefSeq database (accessed in Aug. 2019) and analyzed by antiSMASH 5.0 with default parameters. Lanthipeptide BGCs and precursors were detected automatically by antiSMASH 5.0. Lanthipeptide BGCs with a protein containing both LANC_Like domain (PF05147) and the Pkinase domain (PF00069) were classified as class III or class IV BGCs and further distinguished by profile-Hidden Markov models (pHMMs) developed previously[6].

Proteases in lanthipeptide BGCs were obtained by searching for keywords: "peptidase", "proteinase", "protease", "hydrolase", "beta-lactamase" from antiSMASH generated annotations. These proteases were further subjected to Pfam domain analysis by hmmsearch[43] (HMMER v3.3) with default parameters against the Pfam-A database[27] (v33.1). Pfam domains with hit score >0 with at least 5 occurrences in all lanthipeptide BGCs were selected and manually curated to generate a pool of 120 Pfam domains. These domains were considered as potential lanthipeptide protease Pfams (Supplementary Data 1). Proteases in bacterial genomes were obtained by searching for proteins in bacterial genomes and selecting proteases that contained at least one Pfam domain from the pool with hit score >0.

**Protease and precursor clustering**. Proteases in bacterial genomes were clustered using MMseqs2 (v11.e1a1c) using the easy-cluster workflow with the following parameters: easy-cluster --min-seq-id 0.45 --single-step-clustering --cluster-mode 1 (using "connected component" mode to cover more remote homologs). Lanthipeptide precursor sequences were clustered by MMseqs2 with the same parameters except for a larger similarity cutoff: easy-cluster --min-seq-id 0.6 --single-step-clustering --cluster-mode 1.

**Correlation network construction**. Number of occurrences of protease groups and precursor groups were counted in each bacterial genome. Spearman correlation coefficient and corresponding one-sided *p*-value were calculated for each protease group to precursor group pairs in each genus (Supplementary Note). *P*-values were further adjusted by Benjamini–Hochberg procedure[44] (false-discovery rate). All statistics were calculated in R (v.3.6). Python (v.3.7), pandas (v.0.24.2) and NumPy (v.1.17.4) were used for data processing. Calculated correlations were

filtered by Spearman correlation coefficient ($\rho$), adjusted $p$-values (pAdj) and number of genomes containing the precursor-protease group pair (I). Networks were visualized by Cytoscape[45] (v.3.8.2), in which the size of node indicates the number of proteases or precursors in a genus, width of edge indicates the correlation strength (Spearman coefficient, $\rho$).

**Transcriptomic analysis.** Transcriptomic data from *Streptomyces albidoflavus* J1074, *Streptomyces coelicolor* A3(2), and *Streptomyces davaonensis* JCM 4913 were downloaded from the NCBI SRA database. SRA runs were filtered by paired library layout, random or cDNA library selection, and average length $\geq 100$. A full list of SRA runs used in this study is shown in Supplementary Data 1. BBMap[46] (v38.90) was used to remove base pairs with low quality as well as adapter sequences and PhiX sequences with the following parameters: qtrim=rl ktrim=r mink=11 trimq=10 minlen=45 corresponding to read quality cutoff of 10 and read length cutoff of 45 bp. rRNA sequences were filtered out using SortMeRNA[47] (v.4.3.1) with default parameters with the smr_v4.3_sensitive_db_rfam_seeds database. Quality controlled transcriptomic data were mapped to corresponding genome sequences by BWA mem[48] (v0.7.17) with default parameters. Mapped reads were counted by featureCounts[49] (v2.0.1) with the following parameters: -M -O --fraction -p --primary --minOverlap 40 -Q 10 (minimum mapping quality score of 10, also counted multi-mapping reads). Transcripts per million (TPM) were calculated for each gene.

To calculate proteases that co-expressed with precursors, TPM of genes that belong to the same protease group (or precursor group) were summed to generate a protease and precursor TPM matrix. The matrix was further trimmed to contain only precursor groups and protease groups that exist in the correlation network calculated by genomic data. Spearman correlation coefficient between precursor groups and protease groups were calculated and one-sided $p$-values were adjusted by Benjamini–Hochberg procedure. Correlations presented in the results of all three strains were taken.

For self-sequenced transcriptomic data from *Paenibacillus polymyxa* ATCC 842, the same quality control and statistical analysis workflow was performed.

**Phylogenetic analysis.** For proteases inside or outside BGCs at the genus *Paenibacillus*, protein sequences of all Prot_176 or Prot_819 within the Pre_24-containing BGCs, or in Pre_24-containing genomes (but outside BGCs) were selected. For proteases in genomes that didn't encode Lanthipeptide BGCs, protein sequences of all these members in Prot_176 or Prot_819 were clustered by CD-HIT[50] (v4.8.1) with default parameters to remove redundant sequences sharing $\geq$ 90% sequence similarity, and representative sequences were randomly selected.

Sequences from proteases inside BGCs, outside BGCs, random representatives, as well as a manually added outgroup (WP_023988223.1) were aligned together using MAFFT[51] (v7.480) with the high accuracy linsi method. The resulting alignment was subjected to phylogenetic tree construction by IQ-TREE[52] (v2.1.3) using substitution model LG + G4. Bootstrap was calculated using the Ultrafast bootstrap[53] algorithm implemented in IQ-TREE with 1000 bootstrap replicates. The resulted tree was re-rooted at the manually added outgroup and visualized with Interactive Tree of Life[54].

**Sequence logos.** Full precursor peptide sequences were aligned by MAFFT (v7.480) using the linsi method and trimmed by trimAl[55] (v1.4.rev22) with the -automated1 option. The resulted alignment was used to generate sequence logos using WebLogo[56] (v3.7.4).

**General materials, reagents, strains, and culture conditions.** Biochemicals and media components for bacterial cultures were purchased from Thermo Fisher Scientific Co. Ltd. (USA) unless otherwise stated. Chemical reagents were purchased from standard commercial sources. Restriction endonucleases were purchased from New England Biolabs, Inc. (USA). PCR amplifications were carried out on an Eppendorf® Mastercycler® Nexus X2 Thermal Cycler (Eppendorf Co., Ltd. Germany) using PrimeSTAR HS DNA polymerase (Takara Biotechnology Co., Ltd. Japan). The E.Z.N.A.® Gel Extraction Kit (Omega Bio-tek, Inc., USA) was used for PCR products purification. The NEBuilder® HiFi DNA Assembly master mix (New England Biolabs, Inc., USA) was applied for Gibson assembly[57]. Oligonucleotide synthesis and DNA sequencing were performed by Eton Bioscience, Inc. (USA). The small ubiquitin-like modified (SUMO)-tag gene was synthesized by Bio Basic, Inc. (USA).

All strains used in this study are listed in Supplementary Table 4. *Paenibacillus thiaminolyticus* NRRL B4156, *Bacillus nakamurai* NRRL B41092, *P. taiwanensis* DSM 18679, *P. polymyxa* ATCC 842 and *B. subtilis* 168 strains were cultured in Tryptic Soy Broth (TSB) at 30 °C. *Catenulispora acidiphila* DSM44928 was grown in ISP2 medium at 28 °C. *E. coli* DH10B strains were grown in Luria-Bertani broth (LB) at 37 °C. *E. coli* BL21(DE3) strains were cultured in Terrific Broth (TB) at 37 °C for general growth and 22 °C for protein expression.

**Deletion of *prot_819* and *prot_176* homologs in the heterologous expression host *B. subtilis* 168.** The *prot_819* and *prot_176* homologs in the genome of *B. subtilis* 168, *ymfF* and *ymfH*, were knocked out using the CRISPR-Cas9 system[58]. The sgRNA Designer tool provided by the Broad Institute[59] was used to check the

*ymfF* and *ymfH* genes and a high-scoring 20-nucleotide (nt) sequence was identified. The candidate sgRNA sequence was synthesized as two complementary oligonucleotides and inserted between the BsaI sites of pJOE8999 to construct pJOE8999.1. Then two 700 bp fragments flanking the target region were amplified by PCR and inserted between the SfiI sites of pJOE8999.1 to generate pJOE8999.2. *B. subtilis* 168 was transformed with pJOE8999.2 and grown on LB agar plates containing 5 μg/mL kanamycin and 0.2% mannose. After incubation at 30 °C for 1 day, colonies were placed on LB plates without antibiotics and incubated at 50 °C. On the next day, they were streaked on LB plates to obtain single colonies at 42 °C. Finally, the colonies were tested again for plasmid loss by transferring single colonies to LB agar plates with kanamycin. Knock-out of the *ymfF-ymfH* sequence in selected colonies was confirmed by colony PCR using the outer primers from the 700 bp homology templates.

**Heterologous expression of the *bcn* BGC.** The *bcn* BGC does not contain *prot_819* and *prot_176* homologs. Thus, the *bcn* BGC and the *bcn-gP1/bcn-gP2* including their native promoters were amplified from the genomic DNA of *B. nakamurai* NRRL B41092 with appropriate overhangs for Gibson assembly using corresponding primer listed in Supplementary Table 5. The vector, pDR111, was linearized by digestion with SalI and SphI. The two PCR products (*bcn* BGC and *bcn-gP1/bcn-gP2*) and the linearized pDR111 were ligated via Gibson assembly to construct the heterologous expression plasmid, pDR111-*bcn* + *bcn-gP1/bcn-gP2*, with the *bcn-gP1/bcn-gP2* being downstream of the *bcn* BGC (Supplementary Table 4 and Supplementary Fig. 6a). The pDR111-*bcn* + *bcn-gP1/bcn-gP2* plasmid was integrated into the chromosome of *B. subtilis* 168[60], which was cultivated for production of bacinapeptins A and B as described below.

**Bacinapeptins production and extraction.** The heterologous expression host *B. subtilis* 168 containing plasmid pDR111-*bcn* + *bcn-gP1/bcn-gP2* and the native strain *B. nakamurai* NRRL B-41092 was individually inoculated in a culture tube containing 2 mL of TSB and shaken at 220 rpm, 30 °C, overnight, as seed culture. The seed culture was inoculated in 30 mL of TSB in 150 mL Ultra Yield flasks (3x), which were subsequently shaken at 220 rpm at 30 °C for 3 days. The bacterial broth was extracted with 10 mL butanol, and the organic phase was evaporated under $N_2$. The extracts were redissolved in MeOH to the concentration of 10 mL/mg for LC-MS analysis.

**Heterologous expression of the *ptt* BGC.** The *ptt* BGC containing *pttP1* and *pttP2* was amplified from the genomic DNA of *P. thiaminolyticus* NRRL B4156 with appropriate overhangs for Gibson assembly. The PCR products was ligated with SalI and SphI digested pDR111 via Gibson assembly to generate plasmid pDR111_*ptt* for the heterologous expression of paenithopeptins (Supplementary Fig. 8a). Also, the *ptt-gP1* and *ptt-gP2* out of the *ptt* BGC were amplified with their native promoter and cloned into pDR111 with other *ptt* BGC genes except *pttP1* and *pttP2*, generating plasmid pDR111-*pttΔpttP1/pttP2*+*ptt-gP1/ptt-gP2* (Supplementary Fig. 8b). To construct this plasmid, three fragments were PCR amplified: *ptt-gP1/ptt-gP2*, *pttA1*, and a region containing *pttKC*, *pttA2-pttA7*, *pttMT* and *pttT*. These three PCR products were assembled with SalI and SphI digested pDR111 via Gibson assembly. Additionally, different combinations of *pttA1*, *pttKC*, *pttP1*, and *pttP2*, were constructed into a series of pDR111-based integrative plasmids using the same Gibson assembly procedure as constructing the plasmid pDR111_*ptt* (Supplementary Table 4). In all heterologous expression plasmids for expressing paenithopeptins, native promoters of genes were included. All these plasmids constructed above were individually integrated into the chromosome of *B. subtilis* 168 or *B. subtilis* 168Δ*ymf*FH strain (Supplementary Table 4). The strains harboring these plasmids were cultured, and the culture broth were extracted and analyzed using the same method mentioned above for production and analysis of bacinapeptins.

**Paenithopeptin isolation.** The heterologous expression host *B. subtilis*168 containing pDR111_*ptt* and the native strain *P. thiaminolyticus* NRRL B4156 was individually inoculated in a 150 mL Ultra Yield flask (Thomson Scientific) containing 30 mL of TSB broth and shaken at 220 rpm, 30 °C, overnight, as seed culture. The seed culture was used to inoculate 500 mL of TSB in 2.5 L Ultra Yield flasks (20×) for a total of 10 L TSB, which were subsequently shaken at 220 rpm at 30 °C for 3 days. The bacterial broth was extracted with an equal volume of butanol, and the combined organic phase was concentrated in vacuo to yield the extract, which was partitioned on a reversed-phase C18 open column with a 25% stepwise gradient elution from 50% $H_2O$/MeOH to 100% MeOH. Pure MeOH with 1% formic acid was applied for the final elution. The MeOH/formic acid fraction was further purified by HPLC (Thermo Dionex Ultimate 3000 HPLC system with Chromeleon 7.2.10) on a Phenomenex Luna RP-C18 column (250 mm × 10 mm, 5 μm, 100 Å), with 3.5 mL/min isocratic elution at 32% $H_2O$/MeCN over 30 min with constant 0.1% formic acid) to yield paenithopeptin A. The paenithopeptin A characterization data can be found in the Supplementary Information.

**High-resolution ESI-MS, MSⁿ, and NMR characterization of paenithopeptins and bacinapeptins.** High-resolution ESI-MS spectra and MS$^n$ analysis of paenithopeptins A–E and bacinapeptins A and B were recorded on a Thermo Scientific

Q-Exactive HF-X hybrid Quadrupole-Orbitrap mass spectrometer using electrospray ionization in positive mode. Liquid chromatography was performed on a Thermo Vanquish LC interfaced to the aforementioned mass spectrometer. LC column was a Thermo ProSwift RP-4H with dimensions 1 × 250 mm. Solvent A was 0.1% formic acid in water and Solvent B was 0.1% formic acid in acetonitrile, with the column flow rate being 200 μl/min. The LC gradient started at 10% B for 1 min then went to 100% B by 10 min where it remained for 5 min. MS1 scans were obtained in the orbitrap analyzer which was scanned from 500 to 2000 m/z at a resolution of 60,000 (at 200 m/z). For tandem mass spectrometry (MS2), the relevant parent ion was selected with a 2 m/z window and fragmented it in the HCD cell (collision induced dissociation), using a normalized collision energies of 20, 25 & 30 ev (combined into one spectrum). Fragment ions were then sent to the orbitrap for mass analysis at 30,000 resolution. The MS data was analyzed by Thermo Xcalibur (4.2.47). $^{1}$H, $^{13}$C, $^{1}$H-$^{1}$H COSY, $^{1}$H-$^{1}$H TOCSY, $^{1}$H-$^{1}$H NOESY, $^{1}$H-$^{13}$C HSQC, and $^{1}$H-$^{13}$C HMBC NMR spectra for paenithopeptin A in DMSO-$d_6$ were acquired on a Bruker Avance III HD 400 MHz spectrometer with a 5 mm BBO $^{1}$H/$^{19}$F-BB-Z-Gradient prodigy cryoprobe, a Bruker Avance III HD 500 MHz spectrometer with a PA BBO 500S2 BBF-H-D_05 Z SP probe, or a Bruker Avance III HD Ascend 700 MHz equipped with 5 mm triple-resonance Observe (TXO) cryoprobe with Z-gradients. Data were collected and reported as follows: chemical shift, integration multiplicity (s, singlet; d, doublet; t, triplet; m, multiplet), coupling constant. Chemical shifts were reported using the DMSO-$d_6$ resonance as the internal standard for $^{1}$H-NMR DMSO-$d_6$: δ = 2.50 p.p.m. and $^{13}$C-NMR DMSO-$d_6$: δ = 39.6 p.p.m. The NMR data were processed by MestReNova v12.0.0-20080.

**Determination of absolute configuration of paenithopeptin A.** Paenithopeptin A (1.0 mg) was hydrolyzed in 6 M HCl (700 μL) at 115 °C for 10 h with stirring in a sealed thick-walled reaction vessel, after which the hydrolysate was concentrated to dryness under N$_2$ gas. The resulting hydrolysate was resuspended in distilled H$_2$O (700 μL) and dried again. This process was repeated three times to remove the acid completely. The hydrolysate was divided into two portions (2 × 500 μg) for chemical derivatization with 1-fluoro-2,4-dinitrophenyl-5-L-alanine amide (L-FDAA) and 1-fluoro-2,4-dinitrophenyl-5-D-alanine amide (D-FDAA). Each hydrolysate sample was treated with 1 M NaHCO$_3$ (100 μL) and either L- or D-FDAA (100 μL, 1% solution in acetone) at 40 °C for 1 h. The reaction was then neutralized with addition of 1 M HCl (150 μL) and diluted with MeCN (150 μL). The final sample was analyzed by HPLC-DAD-ESIMS (Kinetex C18 HPLC column, 4.6 × 100 mm, 2.4 μm, 100 Å, 1.0 mL/min gradient elution from 95 to 38% H$_2$O/MeCN over 30 min with constant 0.1% formic acid; positive and negative ionization modes; UV at 340 nm). For the preparation of FDAA derivatives of amino acid standards, 50 μL of each amino acid (L-Ala, L-Ile, L-Val, L-Leu, L-Tyr, L-Lys) (50 mM in H$_2$O) reacted with L- or D-FDAA (100 μL, 1% solution in acetone) at 40 °C for 1 h in the presence of 1 M NaHCO$_3$ (20 μL). The reaction was quenched by 1 M HCl (20 μL) and diluted with MeCN (810 μL) followed by HPLC-DAD-MS analysis using the same column and elution condition as above. The L- and D-FDAA derivatives were detected by either UV or extracted ion chromatograms. Absolute configurations of amino acid residues in paenithopeptin A were established by comparing the retention times of FDAA-derivatized peptide hydrolysate with those of amino acid standards.

**Plasmids construction for in vitro enzymatic assays.** All genes encoding precursor peptides involved in this research were PCR amplified from corresponding genomic DNA with appropriate overhangs for Gibson assembly using primer listed in Supplementary Table 5. The SUMO-tag gene was synthesized with designed overhangs for Gibson assembly. The vector, pET-28a(+), was linearized by digestion with NdeI and EcoRI. Then, the linearized vector, the SUMO-tag gene, and the PCR amplified gene encoding a precursor peptide were ligated via Gibson assembly to generate the corresponding plasmid, with the precursor gene downstream of the SUMO-tag gene, for the expression of His$_6$-SUMO tagged precursor peptide.

To construct plasmids for the expression of lanthipeptide modification enzymes, genes encoding modification enzymes were PCR amplified from corresponding genomic DNA with appropriate overhangs for Gibson assembly using primer listed in Supplementary Table 5. The vector, pHis8, was linearized by digestion with NcoI and EcoRI. The linearized vector and the PCR amplified products were ligated via Gibson assembly to construct the plasmids for the expression of target proteins with an 8xHis tag at the N-terminus.

Mutagenesis in plasmid pHis8-pttP1 and pHis8-pttP2 was performed by PCR-based site-directed mutagenesis[61]. Briefly, primers incorporating the desired base changes were designed and applied through PCR to amplify target genes containing desired mutations. Mutated PCR products were used for Gibson assembly with linearized pHis8 as mentioned above to generate plasmids containing mutations.

**Protein expression and purification.** Plasmid constructed with pHis8 or pET-28(a)+ were transferred into *E. coli* BL21(DE3) by electroporation for protein expression. Precursor peptides were produced in a form tagged with 6 × His fused and a SUMO fusion partner at the N-terminus. Modification proteins were expressed with an 8 × His tag at the N-terminus. *E. coli* BL21(DE3) cells were transformed with pHis8 or pET28 derivative plasmids containing genes encoding

precursor peptides or modification enzymes (Supplementary Table 4). A single colony was used to inoculate a 10 mL culture of LB supplemented with 50 mg/L kanamycin. The culture was grown at 37 °C for 8 h and used to inoculate 1 L of LB with kanamycin. Cells were grown at 37 °C to OD600 ~0.6–0.8, then cooled to 16 °C before IPTG was added to a final concentration of 0.25 mM. The culture was incubated at 20 °C for an additional 12 h. Cells were harvested by centrifugation at 5000 × g for 30 min at 4 °C. Cell pellets were resuspended in 30 mL of lysis buffer (20 mM Tris, pH 8.0, 300 mM NaCl, 25 mM imidazole, 5% glycerol) and the suspension was sonicated on ice for 20 min to lyse the cells. Cell debris was removed by centrifugation at 15,000 × g for 60 min at 4 °C. The supernatant was loaded onto a 3 ml HisSpinTrap™ column (GE Healthcare) previously charged with Ni$^{2+}$ and equilibrated in lysis buffer. The column was washed with 10 mL of wash buffer I (35 mM imidazole, 20 mM Tris, pH 8.0, 300 mM NaCl) and 10 ml of wash buffer II (55 mM imidazole, 20 mM Tris, pH 8.0, 300 mM NaCl). The protein was eluted stepwise with elution buffer I (250 mM imidazole, 20 mM Tris, pH 8.0, 300 mM NaCl) and elution buffer II (500 mM imidazole, 20 mM Tris, pH 8.0, 300 mM NaCl). Resulting elution fractions were collected and analyzed by SDS-PAGE. Fractions containing target proteins were combined and concentrated using an Amicon Ultra-15 Centrifugal Filter Unit (10 kDa for precursor peptide, 30 kDa for modification enzymes, MWCO, Millipore). The resulting protein sample was stored at −70 °C.

**Pulldown assay.** The interaction between Bcn-gP1 and Bcn-gP2 was studied by a pull-down assay using HisSpinTrap™ columns (GE Healthcare) with a bed volume of 200 μl. Binding buffer (20 mM Tris-HCl, pH 8.0, 300 mM NaCl, 5% (v/v) glycerol) was used to immobilize protein (75 nmol) on the column then the column was washed with wash buffer (binding buffer with 30 mM imidazole added) and eluted by elution buffer (binding buffer with 250 mM imidazole added). Bcn-gP1 and His$_8$-Bcn-gP2 were mixed in a ratio of 1:1 and incubated at 30 °C for 2hrs to induce complex formation before applying to the affinity nickel column. A column control was also run to identify and eliminate false positives caused by nonspecific binding of Bcn-gP1, as well as a control with His$_8$-Bcn-gP2 but no Bcn-gP1. We also performed the pull-down assay to explore the interaction between PttP1 and PttP2 using the same procedure. All pull-down assays were repeated three times independently.

**In vitro enzymatic assays.** Assays were based on previous reports on class I and III lanthipeptide enzymes[19,20,29,62]. The class I precursor peptide was treated as previously described for the dehydration of nisin with some modification. Briefly, precursor peptide (pllA; 20 μM) was incubated with cell extracts obtained from BL21 (DE3) expressing pllB and pllC (450 μL each) in 285 μL reaction buffer (100 mM HEPES pH 7.5, 1 mM dithiothreitol (DTT), 10 mM L-glutamate, 10 mM MgCl$_2$, 10 mM KCl, 5 mM ATP) in a final volume of 1.5 mL. The assay was incubated at 30 °C for 5 h after which Tris-HCl pH 8.0 was added to 50 mM and Prot_686 to 2 uM followed by another 2 h incubation at 30 °C. The reaction was quenched and extracted with butanol. The butanol phase was dried under N$_2$, and redissolved in 50 μL 50% methanol/H$_2$O for further LC-ESI-MS analysis.

Class III precursor peptides (100 μM) were incubated with corresponding modification enzymes (20 μM) in 200 μL reaction buffer (20 mM Tris, PH 8.0, 10 mM MgCl$_2$, 1 mM DTT, 5 mM ATP). After 4 h incubation at 30 °C, Prot_176 and/or Prot_819 protease(s) (10 μM each) and 2.5 mM ZnSO$_4$ were added into the reaction.

All in vitro assays were repeated three times independently and the statistical analysis was performed by GraphPad Prism 7.00.

**Protease homology modeling.** Homology-based modeling was performed using the SWISS-MODEL platform[63]. The amino acid sequences of Bcn-gP1/Bcn-gP2 and PttP1/PttP2 were individually used to search for crystal structure templates in the SWISS-MODEL repository. Templates were evaluated according to global mean quality estimate (GMQE) as calculated by SWISS-MODEL, sequence identity and similarity, as well as literature investigation to ensure a heterodimer structure was used for the modeling of putative heterodimers of Bcn-gP1/Bcn-gP2 and PttP1/PttP2. Based on these criteria, the M16B metallopeptidase heterodimer from *Sphingomonas* sp. A1 with PDB code 3amj[37] was selected as the template with a GMQE score of 0.6. The two sequences in this dimer contain the corresponding HXXEH and R/Y motifs found in Bcn-gP1/Bcn-gP2 and PttP1/PttP2, further supporting their use as templates. Models were then visualized and RMSD calculations were performed using PyMol.

## Data availability

161,954 RefSeq genomes used in this research were obtained from NCBI Assembly RefSeq database (https://www.ncbi.nlm.nih.gov/assembly), with a full list of accession numbers provided in Supplementary Data 1. Transcriptomic data for *Streptomyces* were obtained from NCBI SRA database (https://www.ncbi.nlm.nih.gov/sra), with a full list of 80 SRA run accessions provided in Supplementary Data 1. 29,489 lanthipeptide precursor sequences and their clustering information are provided in Supplementary Data 1. 10,263 protease protein IDs are provided in Supplementary Data 1, with all protease sequences, their clustering information, and editable Cytoscape files for Figs. 1c and 2c as well as Supplementary Fig. 3 being deposited in Zenodo (https://zenodo.org/

record/5162875). The crystal structure of Sph2681/Sph2682 used as the template for homology modeling was obtained from Protein Data Bank (PDB code 3amj). Paenilan transcriptomic data are uploaded to NCBI (BioProject PRJNA777777). Data is available from the corresponding authors upon request.

## Code availability

Codes used for computing correlation networks are available on GitHub (https://github.com/yxllab-hku/correlational-network) and Zenodo (https://zenodo.org/record/5842713)[64].

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

## Acknowledgements

J.L. acknowledges the National Institutes of Health (NIH) grants P20GM103641 and P20GM109091, as well as a National Science Foundation EPSCoR Program OIA-1655740. Y.-X.L. acknowledges a Hong Kong Research Grants Council ECS grant HKU27107320 and the Hong Kong Branch of Southern Marine Science and Engineering Guangdong Laboratory (Guangzhou) (SMSEGL20SC02). M.D.W. acknowledges the support of NSF-MRI program (Award No. 1828059) for the acquisition of MS data with the Thermo Q-Exactive mass spectrometer. We thank H.L. and R.H.A. for help with experiments, and T.d.R., J.R.C., P.J.P., T.J., W.E.C., and M.C. for technical advice and access to equipment.

## Author contributions

D.X., Z.Z., Y.-X.L., and J.L. designed research; D.X., E.A.O., Z.Z., N.C., N.D., and L.H. performed research; Z.S. characterized the structures of compounds; M.D.W. performed the LC-MS analysis. D.X., E.A.O., Z.Z., Z.S., N.C., N.D., L.H, P.C., M.D.W., S.-H.D., X.T., H.C., P.N., M.N., Y.-X.L., and J.L. analyzed data and discussed the results. All authors participated in preparing the manuscript.

## Competing interests

The authors declare no competing interests.
