## [Peer Review File · Nature Communications]

REVIEWER COMMENTS

Reviewer #1 (Remarks to the Author):

Xue, Older and co-workers describe an elegant method to discover out-of-cluster biosynthetic enzymes. Most researchers in microbial genome mining of natural products have come across the absence of certain key biosynthetic enzymes in biosynthetic gene clusters (BGCs). There is great value in the field for an effective bioinformatic method that can help identify out-of-cluster enzymes, and there is no doubt that the correlation networks herein described will be of great use. In this paper, the authors focus on out-of-cluster proteases in type-III lanthipeptide biosynthetic gene clusters (BGCs), given about half of BGCs in this class lack in-pathway proteases. The bioinformatic results on the correlation networks are very well done and highlight the multiple applications of this method in the study of BGCs. Further, the discovery and characterization of bacinaeptins and their out-of-pathway proteases validate the findings of the correlational networks. It is essential to provide empirical evidence to support data generated bioinformatically, and the authors have done extensive and great work to do so.

There are, however, some problems in the story flow, especially in the experimental section, which detract from the development of the method and the main point of the manuscript. Whereas bacinaeptins are the perfect case study to highlight correlational networks and identify out-of-BGC biosynthetic enzymes, the excessive focus on the paenithopeptins diverts from the central point of the manuscript. Even though the in-BGC protease class present in the ptt cluster is novel and related to out-of-BGC proteases, these proteases should not be the main focus of the study. In the comments below, there are some suggestions on how to re-write the manuscript to better convey the central point of the paper.

Comments by section:

“Identification of a cryptic protease for maturation of....”

In this section, the authors describe a proof-of concept experiment for their correlation network. In it, 34 proteases are shown to be correlated with Pre_49 class of precursors. What is then done is a transcriptomic analysis to reduce the number of correlations, as it was shown in the previous section. This then leads to the question of whether the correlation network is important at all, if the transcriptomic data can provide all the necessary clues as to which proteases are co-transcribed with precursor peptides. Does the transcriptomic data result in only four classes being co-transcribed? Or do the authors arrive at these four protease classes by looking at the intersection of proteases that appear in the correlation network *and* in the transcriptomic data? This point needs to be clearly stated so that the reader understands the value of the correlation networks.

“Networking analysis reveals a new class...”

Bacinaeptin A and B characterization could be much improved. HRMS measurements are missing for both bacinaeptins, and only low-resolution measurements are provided (HRMS measurements are however provided for other compounds). As a result of the low-resolution measurement, the isotopic patterns for both bacinaeptins are not peptide-like. The MS/MS fragmentation could be improved for bacinaeptin B, where some key ions are missing on the N terminus of the peptide. Which of the proteases (Prot_176 or Prot_819, both?) is responsible for cleavage of the leader peptide? The authors later mention PttP1 and P2 are heteromeric, but discussion of experiments with either one protease seem to be missing. How were bacinaeptins isolated from heterologous expression experiments? It is important to describe whether precursors were tagged for affinity chromatography, how modified peptides were isolated, etc. What are the different bcn cluster arrangement of pDR111 which were expressed in Bacillus?

Much more detail is needed on the description of heterologous expression of the ptt cluster which lead to the discovery of paenitheptins. Importantly, the presence of affinity tags, isolation method and details on the expression construct (i.e. what is the order of genes in constructs? If order of genes was changed, which intergenic regions, if any, were used? Detailed descriptions, along with plasmid maps, would be ideal). Another essential aspect to mention is whether the in-BGC or out-of-BGC proteases used in heterologous expression. Ideally, both in- and out-of-BGC proteases should have been tested, and the results should be clearly stated.

As a broader point in this section: why use the in-BGC proteases from the ptt cluster for in vitro characterization? This experiment does not add to the message of the paper. What about experiments using the out-BGC proteases? The same experiment set should be performed with out-of-cluster proteases. If the point of the paper is to characterize out-of-cluster proteases, why choose this example which contains in-cluster proteases? If about 50% of type-III lanthipeptide BGCs contain out-of-BGC proteases, it is unclear why the authors focus so much on an example that does not support their thesis.

“Different efficiencies of Prot_819/Prot_176...”

Regarding the experiment described in Figure 5: in this case, the paenitheptin product that is being assayed against the proteases contains only the labionin ring. It is possible that the out-of-BGC proteases Ptt-gP1/gP2 could have better activity against fully modified paenitheptins (containing the additional disulfide bridge). Have the authors tested this substrate against the proteases as well?

There is another broader point to be made regarding the main theme of the paper and the flow of the story. The experiments described in Fig. 5 show that the out-of-cluster proteases do not work nearly as well in cleaving the leader peptide of modified paenitheptin and this goes against the core message of the paper. From the outset, the point seems to be to characterize out-of-cluster protease examples, which is nicely done for bacinaeptin A'. The ptt example goes against this central message.

However, the main message of the paper can be reworked to something along the lines of what is written in the last paragraph of the section “Different efficiencies of Prot_819/Prot_176...”. In this section, the authors offer comparisons between pathway-specific and out-of-BGC proteases, and argue that characterization of certain in-pathway proteases can lead to the discovery of out-of-BGC proteases. The story could be written with this message in mind, which would result in describing first the paenitheptin results (in-pathway proteases), and then extend them to the out-of-BGC protease examples in bacinaeptin, as supported by the correlational networks. The story needs to climax at the discovery of out-of-BGC proteases, as is expected from reading the abstract, the introduction and the correlation network method development. However, at the moment, the main focus is in studying the PttP1/P2, which are in-pathway, and the manuscript reads as though it were two separate papers. It would be fine to then use the enzymology studies on PttP1/P2 to further characterize this new enzyme class that appears to be out-of-BGC for so many type III lanthipeptide clusters in Firmicutes.

Other points:

The naming of out-of-pathway biosynthetic enzymes as “cryptic” might generate some confusion in the field. Traditionally, “cryptic” refers to clusters that are not expressed/generate final natural products under standard laboratory conditions. In this case, the out-of-pathway proteases seem to be transcribed and active.

Fig. 4c/d: it is unclear whether the figure shows EICs? UV traces?

MSMS data are missing for compounds depicted in supplementary figures 37, 38, 39, 40.

Second sentence of "Networking analysis reveals a new class...": in "...challenge toward exploiting these lanthipeptides." It would be ideal to elaborate on this point, particularly on how finding these proteases could help you "exploit" class-III lanthipeptides. This would elevate the study to practical utility rather than a proof-of-concept for the correlational networks.

Along the lines of exploiting these results, have the authors attempted to find biological activities for the isolated compounds?

Reviewer #2 (Remarks to the Author):

This is a very elegant study by Jie Li and colleagues that uses a network analysis approach to identify essential genes involved in natural products biosynthesis that are located outside of the biosynthetic operon. In particular, the analysis focuses on a class of ribosomal synthesized and post-translationally modified peptides (RiPPs) termed lanthipeptides because of the occurrence of lanthionine residues that are installed post-translationally. Often in the biosynthetic gene clusters (BGCs) of lanthipeptides (and other classes of RiPPs), the leader protease that serves to elaborate the final product is not located in the biosynthetic cluster. Here, the authors use the networking approach and complementary biochemical experiments to identify proteases that don't share synteny with the BGC. The work is well carried out and should be of significance to researchers interested in natural products, pathway engineering, and synthetic biology. The manuscript is well above the bar for publication in Nat. Comms. but I would ask for some editorial changes in the manuscript prior to acceptance.

1. The author state that there are 4 classes of lanthipeptide syntheses. This is no longer true as the cacoidin BGC represent a fifth class of lanthipeptide synthetases. I realize that it may be too much to ask for their analysis to extend to the class V enzymes but the authors should acknowledge the existence of this class.

2. For non-specialists, could the authors add a sentence about why they chose Spearman rank order correlation vs. other statistical methods.

3. Figure 2, panel d; please state what EIC stands for in the figure legend (extracted ion chromatogram).

4. Figure 3. I realize that space is limited but it would really be useful to include the structure of paenitheptin A somewhere in this figure.

5. Figure 4, panel b is not particularly instructive since PttP1 and PttP2 can't be easily distinguished by SDS-PAGE given their similar masses. Can the authors provide something that is clearer evidence? Perhaps urea denature the complex and show that P2 elutes in the wash and His-P1 is still on the beads?

In short, this manuscript provides meaningful analysis into a long-standing mystery in the natural products biosynthesis community. I heartily endorse consideration.

Reviewer #3 (Remarks to the Author):

In their manuscript, Xue et al describe an in silico (based on Spearman' rank-order correlation) approach to identify proteases required to process and activate lanthipeptide natural products that are not encoded within the biosynthetic gene clusters encoding their production. They

validated their approach by in silico identifying lanthipeptide proteases with previous experimental knowledge of their involvement in their correlation network. As the main application of their method, the authors identified and in vitro characterised the orphan protease involved in paenilan maturation, and, most interestingly, identified and extensively characterised two members of the M16B metalloprotease family, which was not associated with lanthipeptide biosynthesis before, and described the corresponding new lanthipeptides (bacinapeptins, paenithopeptins).

This is a highly interesting and very comprehensive study addressing one of the enigmas of "Ribosomal and post-translational modified peptide (RiPP)" research and therefore will be of great interest for scientists interested in computational genome mining for BGCs and also experimentalists working on elucidating RiPP biosynthesis pathways and their application.

The paper is well written and the authors managed to nicely present the huge amount of methods and data ranging from in silico analyses, protein expression/purification and in vitro assays, lanthipeptide structure elucidation (MS and NMR-based) transcriptomics/co-expression analysis (on existing data), mutagenesis / in vivo studies, molecular modeling. The data is analysed and interpreted to a high standard and combined to provide a strong support for the main conclusions of the manuscript.

There are a few issues, which should be addressed before publication:

* **Title:** The title suggests that the author's correlation-networking method is generally applicable to identify genes encoding essential biosynthetic enzymes that are not encoded within the BGCs. The authors impressively and convincingly demonstrate its use in identifying "orphan" proteases involved in RiPP biosynthesis pathways. However, in this case the target space of genes/enzymes to search for is very limited - it is known that a protease needs to be involved. And even with this prior knowledge, the search space included more than 23 million protease sequences. Is this really applicable in cases where this prior knowledge is not existing (like in the majority of cases for novel BGCs with exception of the RiPPs)? To justify the broad title, I would expect further examples (for example on the other examples of BGCs/pathways that involve "cryptic" enzymes that are mentioned in the introduction). If this is not possible I would recommend to modify the title for the application in finding orphan proteases also in the title.

* **To reproduce the in silico workflow** , the scripts used for analysing the data should be made available.

* **Legend Figure 2 panel B:** The authors should mention the source (species, accession) of this gene clusters, as otherwise in my opinion specifying the exact distance between BGC and protease makes not much sense.

* **Page 10 (Bcn-gP1/2 // PttP1/2 characterisation):** The authors state that both subunits need to be present for efficient proteolysis - however the HPLC diagrams show that the are also correct products formed with only subunit 2 (Bcn-gP2 or PttP2) respectively - this should be better explained.

* **Page 11 / Figure 5:** I'm surprised that PttP1/2 show higher activity towards BccnA1 than the "native" protease Bcn-gP1/2

* **Is there any information available on the bioactivities of bacinapeptins and paenithopeptins?**

Authors' Responses to Reviewer comments

Reviewer #1 (Remarks to the Author):

1. Xue, Older and co-workers describe an elegant method to discover out-of-cluster biosynthetic enzymes. Most researchers in microbial genome mining of natural products have come across the absence of certain key biosynthetic enzymes in biosynthetic gene clusters (BGCs). There is great value in the field for an effective bioinformatic method that can help identify out-of-cluster enzymes, and there is no doubt that the correlation networks herein described will be of great use. In this paper, the authors focus on out-of-cluster proteases in type-III lanthipeptide biosynthetic gene clusters (BGCs), given about half of BGCs in this class lack in-pathway proteases. The bioinformatic results on the correlation networks are very well done and highlight the multiple applications of this method in the study of BGCs. Further, the discovery and characterization of bacinapeptins and their out-of-pathway proteases validate the findings of the correlational networks. It is essential to provide empirical evidence to support data generated bioinformatically, and the authors have done extensive and great work to do so.

There are, however, some problems in the story flow, especially in the experimental section, which detract from the development of the method and the main point of the manuscript. Whereas bacinapeptins are the perfect case study to highlight correlational networks and identify out-of-BGC biosynthetic enzymes, the excessive focus on the paenithopeptins diverts from the central point of the manuscript. Even though the in-BGC protease class present in the ptt cluster is novel and related to out-of-BGC proteases, these proteases should not be the main focus of the study. In the comments below, there are some suggestions on how to re-write the manuscript to better convey the central point of the paper.

Response 1: We appreciate both the positive comments and the constructive advice from this reviewer. Please see below our point-by-point responses that we believe have addressed all the comments.

Comments by section:

“Identification of a cryptic protease for maturation of...”

2. In this section, the authors describe a proof-of concept experiment for their correlation network. In it, 34 proteases are shown to be correlated with Pre_49 class of precursors. What is then done is a transcriptomic analysis to reduce the number of correlations, as it was shown in the previous section. This then leads to the question of whether the correlation network is important at all, if the transcriptomic data can provide all the necessary clues as to which proteases are co-transcribed with precursor peptides. Does the transcriptomic data result in only four classes being co-transcribed? Or do the authors arrive at these four protease classes by looking at the intersection of proteases that appear in the correlation network *and* in the transcriptomic data? This point needs to be clearly stated so that the reader understands the value of the correlation networks.

Response 2: We totally agree with this comment. In fact, the 4 proteases groups were revealed by looking for the intersection of the 34 proteases identified from correlation network and the transcriptomic data. This point is now clearly stated in the revised manuscript, as shown by the revisions highlighted in red in the following parts: Page 5, lines 154-157; Page 7, lines 217-220; and the legend of Figure 2 (Page 6).

Regarding “whether the correlation network is important”, besides what is mentioned above, we would like to clarify a bit more: below we will first articulate the importance of correlational networking analysis

at the genomic level, and then explain why we integrated co-expression analysis to complement correlational analysis.

Importance of the correlational networking analysis: many lanthipeptide precursors were strongly correlated with only one or two groups of proteases in our correlational networking at the genus level. For example, for class III lanthipeptides, “among the 91 significant correlations representing 864 precursors, 758 precursors (88%) were strongly correlated with only one or two groups of proteases” (Page 5, lines 133-135). Even for the remaining 12% of the precursors that correlated to multiple groups of proteases, “the correlation strength between different groups of proteases and the same group of precursors appeared to be different based on Spearman’s rank correlation coefficient (ρ). In addition, we noticed that multiple-correlation clusters could be distinguished by the classes of their correlated proteases” (Page 5, lines 149-152). These explanations were stated in the original version of the manuscript and are highlighted in green in the revised manuscript for easier reading. In contrast, only using co-expression analysis is not practical at current stage and also has several difficulties associated: 1. Compared to genomic data, not enough transcriptomic data, especially whole genome transcriptomic data, are currently available online for free access; 2. Big data co-expression analysis would demand much higher requirements for computational capacity, resources, and time; 3. Many genes are also likely transcribed at a certain condition leading to many candidate genes.

Why we integrated co-expression analysis to complement correlational analysis: on one hand, we aimed to even simplify the multiple-correlation clusters in our results, although they only accounted for a small portion of the lanthipeptide precursors; On the other hand, we sought to explore a proof of concept whether integrating available transcriptomic data into our established correlational network can facilitate rapid identification of proteases. Once proven effective, we envision that with increasingly more genomic and transcriptomic data, our platform integrating both could be instrumental in identifying lanthipeptide proteases and potentially expand to identify other unclustered natural product biosynthetic genes.

Therefore, we hope we have shown that integration of co-expression analysis would not conflict with correlational networking analysis at the genomic level, instead, it will complement the correlational network.

“Networking analysis reveals a new class...”

3. Bacinapeptin A and B characterization could be much improved. HRMS measurements are missing for both bacinapeptins, and only low-resolution measurements are provided (HRMS measurements are however provided for other compounds). As a result of the low-resolution measurement, the isotopic patterns for both bacinapeptins are not peptide-like. The MS/MS fragmentation could be improved for bacinapeptin B, where some key ions are missing on the N terminus of the peptide.

Response 3: We appreciate this comment. In the revised manuscript, HRMS spectra of bacinapeptins A and B (Supplementary Figs. 6c and 7b) were used to replace the original spectra. The new HRMS spectra are of much higher quality. In addition, for bacinapeptin B, we acquired new higher quality MS/MS fragmentation with more b-ions on the N-terminus, as shown in Supplementary Fig. 7c.

4. Which of the proteases (Prot_176 or Prot_819, both?) is responsible for cleavage of the leader peptide? The authors later mention PttP1 and P2 are heteromeric, but discussion of experiments with either one protease seem to be missing.

Response 4: In the revised manuscript, we made it clear that “these results indicated Prot_819/Prot_176 as previously unknown lanthipeptide proteases and established their role as a heteromeric complex in the maturation of novel class III lanthipeptides...” (Page 11, line 347). Prot_819 and Prot_176 work together to achieve the maximal efficiency in cleaving leader peptides, because Prot_176 possesses an HXXEH motif essential for Zn²⁺ binding and catalytic activity, while Prot_819 contains an R/Y pair to facilitate substrate binding. We later “individually mutated H67, E70, and H71 of the HXXEH motif and R298 and Y305 of the R/Y pair to Ala, leading to decreased activity... (Supplementary Fig. 31). Simultaneous mutation of all these five residues into Ala residues completely abolished the production... (Supplementary Fig. 31)” (Page 12, lines 403-407).

In the original manuscript, we pointed out that Bcn-gP1 and PttP1 belong to Prot_819, while Bcn-gP2 and PttP2 belong to Prot_176, respectively (Page 7, lines 253-256 and Page 9, lines 289-292). We also defined that the writing of Bcn-gP1/Bcn-gP2, PttP1/PttP2, and Prot_819/Prot_176 (with a slash in between two proteins) means the two proteins form a complex (Page 10, lines 318-319). Later we described that “...by adding Bcn-gP1, Bcn-gP2, or both, respectively... We observed that (i) Bcn-gP1 and Bcn-gP2 were simultaneously required to generate the highest yield of the product...; and (ii) the Bcn-gP1/Bcn-gP2 complex has specificity against BcnKC-modified precursor bearing a labionin ring (Fig. 4c). Likewise, we performed an in vitro characterization of PttP1/PttP2... We observed consistent results (Fig. 4d) as described for Bcn-gP1/Bcn-gP2” (Page 11, lines 328-336).

5. How were bacinapeptins isolated from heterologous expression experiments? It is important to describe whether precursors were tagged for affinity chromatography, how modified peptides were isolated, etc. What are the different bcn cluster arrangement of pDR111 which were expressed in Bacillus?

Response 5: We appreciate this comment. We made corresponding changes (highlighted in red) in the main text of the revised manuscript (Page 7, lines 257-258). Furthermore, in the “Methods” section of the revised manuscript, we added a new subsection called “Heterologous expression of the *bcn* BGCs” (Page 20, lines 699-707). In this subsection, we describe in detail the construction of the *bcn* BGC into the pDR111 vector, the cultivation of the heterologous expression host, and the production and extraction of bacinapeptins from the host. Specifically, we cloned the *bcn* BGC into pDR111 with each gene’s native promoter and the same gene organization as in the native strain *B. nakamurai* NRRL B-41092. Since the *bcn* BGC does not harbor any protease-encoding genes, we also cloned from the genome of the native strain a pair of the potential protease genes, *bcn-gP1/bcn-gP2*, identified by our correlational networking and put them downstream of the *bcn* BGC in pDR111. The precursors of the *bcn* BGC were not tagged for purification, instead, bacinapeptins with molecular masses around 2000 Da were produced as free small peptide molecules, extracted by butanol, pre-fractionated on a small reversed-phase column followed by LC-MS analysis.

To complement this newly added subsection, the plasmid map containing the *bcn* BGC along with protease genes was added in Supplementary Figs. 6a. Supplementary Table 4 also lists the relevant plasmid information.

6. Much more detail is needed on the description of heterologous expression of the ptt cluster which lead to the discovery of paenithopeptins. Importantly, the presence of affinity tags, isolation method and details on the expression construct (i.e. what is the order of genes in constructs? If order of genes was changed, which intergenic regions, if any, were used? Detailed descriptions, along with plasmid maps,

would be ideal). Another essential aspect to mention is whether the in-BGC or out-of-BGC proteases used in heterologous expression. Ideally, both in- and out-of-BGC proteases should have been tested, and the results should be clearly stated.

Response 6: We appreciate this comment. We made corresponding changes (highlighted in red) in the main text of the revised manuscript (Page 9, lines 293-296). Furthermore, in the “Methods” section of the revised manuscript, we added a subsection called “Heterologous expression of the *ptt* BGCs” (Page 20, lines 717-732). In this newly added subsection, we describe in detail the construction of the *ptt* BGC into the pDR111 vector, the cultivation of the heterologous expression host, and the production of paenitheptins from the host. Specifically, the *ptt* BGC was cloned into pDR111 with the native promoter of each gene, and the *ptt* BGC in the plasmid remained the same gene organization as in the native strain *P. thiaminolyticus* NRRL B-4156. The precursors of the *ptt* BGC in the plasmid were not tagged for purification, instead, paenitheptins with molecular masses around 2000 Da were produced as free small peptide molecules, extracted by butanol, and purified using reversed-phase C18 open column and HPLC.

As suggested by this reviewer, besides cloning the intact *ptt* BGC including the in-BGC protease-encoding genes *pttP1* and *pttP2* as described above, we also used the out-of-BGC protease genes *ptt-gP1* and *ptt-gP2* in replacement of *pttP1* and *pttP2* and generated another pDR111-based plasmid for comparative heterologous expression. The results showed that the *ptt* BGC along with the in-BGC protease genes led to higher production yield than that of the *ptt* BGC with the out-of-BGC protease genes. Following this reviewer’s comment, these results were clearly stated in the main text (Page 9, lines 293-296) and shown in Supplementary Fig. 8c of the revised manuscript.

To complement this newly added subsection, the plasmid maps containing the *ptt* BGC along with either the in-BGC or out-of-BGC protease genes were added in Supplementary Figs. 8a and 8b. Supplementary Table 4 also lists the relevant plasmids information.

7. As a broader point in this section: why use the in-BGC proteases from the *ptt* cluster for in vitro characterization? This experiment does not add to the message of the paper. What about experiments using the out-BGC proteases? The same experiment set should be performed with out-of-cluster proteases. If the point of the paper is to characterize out-of-cluster proteases, why choose this example which contains in-cluster proteases? If about 50% of type-III lanthipeptide BGCs contain out-of-BGC proteases, it is unclear why the authors focus so much on an example that does not support their thesis.

Response 7: We very much appreciate this comment that let us reflect on the possible omission or unclarity of the logic and/or information provided in our original manuscript that may result in possible difficulty or misunderstanding for reviewers and readers to follow the story. We added a few transition sentences and made corresponding changes throughout the revised manuscript (highlighted in red), aiming to make our logic and rationale clear regarding why we selected two examples of BGCs along with respective out-of-BGC and in-BGC proteases to study and present the results at the same time in a seemingly parallel organizing in the section “Networking analysis reveals a new family of lanthipeptide proteases...”. We also explain about this in detail in this Response 7 below. Since this comment is related to the next broader point of this reviewer, please read our Response 7 to this comment together with our Responses 8 and 9 addressing the point regarding the main theme of the manuscript and the flow of the story.

A major reason to perform an in vitro enzymatic assay for the pathway-specific protease pair PttP1/PttP2 in this section was to lay a foundation for the enzyme efficiency comparison presented in the next section.

This efficiency comparison included three hidden protease pairs (out-of-BGC) and one in-BGC protease pair that was actually PttP1/PttP2. This efficiency comparison also included Ptt-gP1/Ptt-gP2, the out-of-BGC counterpart of PttP1/PttP2 in the same *ptt* BGC, which can address the reviewer's comment that "the same experiment set should be performed with out-of-cluster proteases". This efficiency comparison section was important, because it led to our evolution hypothesis that "certain proteases with general functions in the genome might have evolved more or less specific activity against class III lanthipeptides, and extra copies of them might have further evolved into pathway-specific proteases for enhanced activity and specificity" (Page 12, lines 387-390). This in turn supports our initial foundation for correlation networking: substrate specificity for proteases. This manuscript not only aims to just show out-of-BGC proteases we identified, but also seeks to complete a closed logic circle about why we can develop this correlational networking and how we can use it to identify proteases. Thus, from this perspective, we believe that it is necessary to include an in-BGC protease as an example.

Then, one may argue why not mentioning Ptt-gP1/Ptt-gP2 in the former section (the protease discovery section) and PttP1/PttP2 in the later section (the efficiency comparison section). This was mainly because: Bcn-gP1/Bcn-gP2 already represented an out-of-BGC protease pair that was selected as the first example in the protease discovery section, and Ptt-gP1/Ptt-gP2 pair is very similar to Bcn-gP1/Bcn-gP2. Instead, using PttP1/PttP2 as the second example in this section would represent an in-BGC protease pair that may be more different from Bcn-gP1/Bcn-gP2, so that both scenarios of out-of-BGC and in-BGC proteases were included in this section. We feel that it was more appropriate to include both out-of-BGC and in-BGC proteases before we verified that these potential proteases are indeed correct proteases, thus laying a better foundation for subsequent efficiency comparison.

Nevertheless, we agree that out-of-BGC proteases are the focus of this manuscript. Thus, we did make several changes/additions in the revised manuscript to emphasize the discovery and characterization of out-of-BGC proteases, including:

- A. The *ptt* BGC, with *ptt-gP1* and *ptt-gP2* replacing *pttP1* and *pttP2*, was also cloned into pDR111 for heterologous expression (Page 9, lines 293-294; Supplementary Fig. 8c).
- B. On Page 9, lines 307-311, we added an explanation why we selected the pairs of Bcn-gP1/Bcn-gP2 and PttP1/PttP2 as examples to study the proteolytic activity.
- C. We added a pull-down assay to study the protein-protein interaction between Bcn-gP1 and Bcn-gP2. The assay result is shown in Figure 4b (Page 10, lines 321-323).
- D. On Page 11, lines 354-363, when describing the protease gene distributions, we reorganized them into three scenarios to better emphasize the out-of-BGC proteases.
- E. We made sure when we need to present results for both Bcn-gP1/Bcn-gP2 and PttP1/PttP2 in the protease discovery/characterization section, the flow of organizing was based on the out-of-BGC Bcn-gP1/Bcn-gP2 (the parts highlighted in green on Pages 10 and 11).
- F. In the section "Member of Prot_819/Prot_176 shows specificity and unique activity...", although PttP1/PttP2 was used due to its highest activity, we specially clarified that PttP1/PttP2 is "a representative member of Prot_819/Prot_176 that are widely distributed outside of many class III lanthipeptide BGCs" (Page 12, lines 401-402).

In summary, our correlational analysis is global and identifies proteases "regardless of being encoded inside or outside of a BGC, with an emphasis on identification of hidden lanthipeptide proteases". Although this manuscript emphasizes out-of-BGC proteases, including an in-BGC protease would lay a foundation for efficiency comparison and complete the logic circle of our correlational analysis. In addition, we made corresponding changes/additions in the revised manuscript to ensure that the out-of-BGC proteases were the focus of this manuscript.

“Different efficiencies of Prot_819/Prot_176...”

8. There is another broader point to be made regarding the main theme of the paper and the flow of the story. The experiments described in Fig. 5 show that the out-of-cluster proteases do not work nearly as well in cleaving the leader peptide of modified paenithopeptin and this goes against the core message of the paper. From the outset, the point seems to be to characterize out-of-cluster protease examples, which is nicely done for bacinapeptin A'. The ptt example goes against this central message.

Response 8: Please kindly see our Response 7 above: in summary, including an in-BGC protease would lay a foundation for enzyme efficiency comparison, leading to our evolution hypothesis regarding presumably evolutionarily gained higher substrate specificity and catalytic activity, which in turn supports our initial foundation for correlation networking to eventually complete a closed logic circle of this manuscript.

In addition, the in-BGC protease of the *ptt* BGC has an out-of-BGC counterpart (*ptt-gP1/ptt-gP2*) that was also studied in this manuscript (Page 9, lines 293-294). Thus, we believe that the *ptt* BGC is a good example to include in this manuscript.

9. However, the main message of the paper can be reworked to something along the lines of what is written in the last paragraph of the section “Different efficiencies of Prot_819/Prot_176...”. In this section, the authors offer comparisons between pathway-specific and out-of-BGC proteases, and argue that characterization of certain in-pathway proteases can lead to the discovery of out-of-BGC proteases. The story could be written with this message in mind, which would result in describing first the paenithopeptin results (in-pathway proteases), and then extend them to the out-of-BGC protease examples in bacinapeptin, as supported by the correlational networks. The story needs to climax at the discovery of out-of-BGC proteases, as is expected from reading the abstract, the introduction and the correlation network method development. However, at the moment, the main focus is in studying the PttP1/P2, which are in-pathway, and the manuscript reads as though it were two separate papers. It would be fine to then use the enzymology studies on PttP1/P2 to further characterize this new enzyme class that appears to be out-of-BGC for so many type III lanthipeptide clusters in Firmicutes.

Response 9: We appreciate such detailed suggestions. As mentioned by Reviewer 3, this manuscript contains a “huge amount of methods and data”. When we were preparing this manuscript, we indeed attempted different ways of organizing the flow of the story, including exactly the same way as suggested by this reviewer. We eventually decided to introduce both Bcn-gP1/Bcn-gP2 and PttP1/PttP2 in the same protease discovery section, then transitioning to the efficiency comparison section and climaxing at the enhanced proteolytic efficiency to close the logic circle of the correlational analysis. This is explained in detail in Response 7 above.

When we tried the flow of the story starting from in-BGC protease and then moving to out-of-BGC protease, it read to us as if a simple blast search based on the in-BGC proteases leads to the discovery of out-of-BGC proteases. We feel this layout may cause some misunderstandings. In fact, our correlational analysis is different from blast search, and it discovers proteases “regardless of being encoded inside or outside of a BGC” (Page 3, lines 95-97), using a collection of “120 Pfam domains to search for proteases from the full set of 161,954 bacterial genomes” (Page 3, lines 109-110). In addition, if we organize the flow of story from in-BGC proteases to out-of-BGC proteases, to efficiency comparison, and then back to in-BGC protease with enhanced activity, we feel that the flow is a bit too long, especially we still have

multiple other sections such as establishment and validation of correlation network, identification of protease for paenilan, and detailed characterization of Prot_819/Prot_176 member.

Finally, although we introduced both Bcn-gP1/Bcn-gP2 and PttP1/PttP2 in the same protease discovery section, we did make several changes/additions in the revised manuscript to emphasize the discovery and characterization of out-of-BGC proteases (please see the second to last paragraph of Response 7).

We have extensively discussed about the story flow among all co-authors and other colleagues, and the current organization received the best feedback like what Reviewer 3 commented, “the paper is well written and the authors managed to nicely present the huge amount of methods and data...”. We understand that even under the current organization, this manuscript likely reads not simple due to its huge number of results. We appreciate the understanding of this reviewer.

10. Regarding the experiment described in Figure 5: in this case, the paenithopeptin product that is being assayed against the proteases contains only the labionin ring. It is possible that the out-of-BGC proteases Ptt-gP1/gP2 could have better activity against fully modified paenithopeptins (containing the additional disulfide bridge). Have the authors tested this substrate against the proteases as well?

Response 10: This is a good point. Since it is not clear if the additional disulfide bridge is introduced before or after the leader peptide is removed, we did not test the substrate with both labionin ring and additional disulfide bridge in the in vitro assay shown in Figure 5. However, to address this from a different angle, in the revised manuscript, we added an in vivo heterologous expression assay involving the out-of-BGC protease gene pair (*ptt-gP1/ptt-gP2*), which was actually suggested by this reviewer’s comment 6. The results showed a similar trend to that of the in vitro assay in Figure 5: *pttP1/pttP2* exhibited higher efficiency than that of *ptt-gP1/ptt-gP2* in producing final mature paenithopeptins with the additional disulfide bridge, as shown in the main text (highlighted in red) of the section “**Networking analysis reveals a new family of lanthipeptide proteases...**” (Page 9, lines 293-296) and Supplementary Fig. 8c. Therefore, we believe that *pttP1/pttP2* possess higher efficiency than that of *ptt-gP1/ptt-gP2* in producing fully modified paenithopeptins containing the additional disulfide bridge.

Other points:

11. The naming of out-of-pathway biosynthetic enzymes as “cryptic” might generate some confusion in the field. Traditionally, “cryptic” refers to clusters that are not expressed/generate final natural products under standard laboratory conditions. In this case, the out-of-pathway proteases seem to be transcribed and active.

Response 11: We agree with this comment. Accordingly, in the revised manuscript, we have changed “cryptic” to “unclustered protease genes/hidden proteases”.

12. Fig. 4c/d: it is unclear whether the figure shows EICs? UV traces?

Response 12: Thanks for pointing this out. It shows EICs, which has been clearly stated in Figure 4c and 4d (Page 10) as well as in Figure 2 (Page 6) in the revised manuscript.

13. MSMS data are missing for compounds depicted in supplementary figures 37, 38, 39, 40.

Response 13: The major products showed in Supplementary Figs. 37-40 are Paenithopeptins A2, A3, A5, and A7, the MSMS data of which were shown in Supplementary Figs. 33-36 in the original manuscript. In addition, in the revised manuscript, we added the MSMS data for the minor peaks in these figures: Supplementary Figs. 37c-e for paenithopeptin A2(-1), paenithopeptin A2(-2), and paenithopeptin A2(-3); Supplementary Figs. 38c-f for paenithopeptin A3(-1), paenithopeptin A3(-2), paenithopeptin A3(-3), and paenithopeptin A3(-4); Supplementary Figs. 39c-e for paenithopeptin A5(-1), paenithopeptin A5(-2), and paenithopeptin A5(-3); and Supplementary Figs. 40c-e for paenithopeptin A7(-1), paenithopeptin A7(-2), and paenithopeptin A7(-3). Furthermore, we added the MSMS data for bacinapeptin A' in Supplementary Fig. 29d.

14. Second sentence of "Networking analysis reveals a new class...": in "...challenge toward exploiting these lanthipeptides." It would be ideal to elaborate on this point, particularly on how finding these proteases could help you "exploit" class-III lanthipeptides. This would elevate the study to practical utility rather than a proof-of-concept for the correlational networks.

Response 14: We agree with this comment. Accordingly, in the revised manuscript, we added a sentence explaining that finding the missing proteases can contribute to "heterologous expression for the discovery of new class III lanthipeptides, pathway bioengineering for increasing production yield, chemical diversity, and biological activities, and leveraging the enzymology of proteases as a synthetic biology tool for general proteolytic and traceless tag removal applications" (Page 7, lines 234-237). In addition, in the original manuscript, we also cited reference 18, a review article that includes protease-related open questions for class III lanthipeptides and provides more detailed information to help readers understand the role and significance of proteases in class III lanthipeptide biosynthesis and application. Since this section is part of the Results section, we feel that we are supposed to still focus on presenting our results in this section. Thus, we hope that our current revision can address this comment.

15. Along the lines of exploiting these results, have the authors attempted to find biological activities for the isolated compounds?

Response 15: The similar comment was also mentioned by Reviewer #3. We totally agree that the biological activities are important. Thus, we indeed have been studying the biological activities for not only the isolated compounds of this manuscript but also other structurally related compounds. Due to the theme and the already extensive data of this manuscript, we think that it would be the best to report the biological activities in a separate paper in a timely manner, which we hope the reviewer agree with.

Reviewer #2 (Remarks to the Author):

This is a very elegant study by Jie Li and colleagues that uses a network analysis approach to identify essential genes involved in natural products biosynthesis that are located outside of the biosynthetic operon. In particular, the analysis focuses on a class of ribosomal synthesized and post-translationally modified peptides (RiPPs) termed lanthipeptides because of the occurrence of lanthionine residues that are installed post-translationally. Often in the biosynthetic gene clusters (BGCs) of lanthipeptides (and other classes of RiPPs), the leader protease that serves to elaborate the final product is not located in the biosynthetic cluster. Here, the authors use the networking approach and complementary biochemical experiments to identify proteases that don't share synteny with the BGC. The work is well carried out and should be of significance to researchers interested in natural products, pathway engineering, and

synthetic biology. The manuscript is well above the bar for publication in Nat. Comms. but I would ask for some editorial changes in the manuscript prior to acceptance.

1. The author state that there are 4 classes of lanthipeptide syntheses. This is no longer true as the cacoidin BGC represent a fifth class of lanthipeptide synthetases. I realize that it may be too much to ask for their analysis to extend to the class V enzymes but the authors should acknowledge the existence of this class.

Response 1: We appreciate this reviewer pointing this out and his/her kind understanding of extensive work at this stage to extend to the class V enzymes. When we initiated our analysis, we had not noticed any reported class V lanthipeptides. Close to the end of our analysis, we began to notice the emerging of sparse class V lanthipeptide cases, but without a defined standard for classification of this new class at that stage. Nevertheless, we followed this reviewer's suggestion to acknowledge the existence of this new class and explained the reason for not including class V enzymes in our analysis of this manuscript (Page 3, lines 82-85).

2. For non-specialists, could the authors add a sentence about why they chose Spearman rank order correlation vs. other statistical methods.

Response 2: In contrast to Pearson's correlation that is also frequently used in correlational analysis, Spearman's correlation accounts for the "ranking" of data instead of exact values and does not require a normal distribution of the data, which matches the gene distribution in our data. This has been added in the revised manuscript (Page 3, lines 97-100).

3. Figure 2, panel d; please state what EIC stands for in the figure legend (extracted ion chromatogram).

Response 3: EIC stands for extracted ion chromatogram, which has been added in the legend of Figure 2 (Page 6).

4. Figure 3. I realize that space is limited but it would really be useful to include the structure of paenithopeptin A somewhere in this figure.

Response 4: We agreed and attempted many different layouts and arrangements of Figure 3 trying to include the chemical structure of paenithopeptin A in this figure, however, we regrettably could not manage to do so due to the really limited space in Figure 3 (the original Figure 3 already occupies an entire page in the manuscript). Nevertheless, in the revised manuscript, the chemical structure of paenithopeptin A has been added in Supplementary Fig. 10b. In addition, we added all chemical structures of bacinaeptins and paenithopeptins that have distinct precursor peptides in Supplementary Figs. 6e, 7d, 33c, 34c, 35c, and 36c.

5. Figure 4, panel b is not particularly instructive since PttP1 and PttP2 can't be easily distinguished by SDS-PAGE given their similar masses. Can the authors provide something that is clearer evidence? Perhaps urea denature the complex and show that P2 elutes in the wash and His-P1 is still on the beads?

Response 5: We performed another pull-down assay for Bcn-gP1 and His-Bcn-gP2 that have larger differences (2.3 KDa) between these two protein masses and thus enabled a clear distinguishing between their bands in the SDS-PAGE gel. This new pull-down result was added as Figure 4b in the revised manuscript (Page 10). As shown in this figure, His₈-tag-free Bcn-gP1 alone was eluted in the flowthrough while His₈-tagged Bcn-gP2 alone was present in the elute. In contrast, His₈-tag-free Bcn-gP1 was

immobilized on the nickel affinity column when it formed a complex with His₈-tagged Bcn-gP2, and then both proteins were eluted in the elute as two bands.

In short, this manuscript provides meaningful analysis into a long-standing mystery in the natural products biosynthesis community. I heartily endorse consideration.

Response 6: We thank this reviewer for the overall high comments and specific constructive advice and suggestions.

Reviewer #3 (Remarks to the Author):

In their manuscript, Xue et al describe an in silico (based on Spearman' rank-order correlation) approach to identify proteases required to process and activate lanthipeptide natural products that are not encoded within the biosynthetic gene clusters encoding their production. They validated their approach by in silico identifying lanthipeptide proteases with previous experimental knowledge of their involvement in their correlation network. As the main application of their method, the authors identified and in vitro characterised the orphan protease involved in paenilan maturation, and, most interestingly, identified and extensively characterised two members of the M16B metalloprotease family, which was not associated with lanthipeptide biosynthesis before, and described the corresponding new lanthipeptides (bacinaeptins, paenithoeptins).

This is a highly interesting and very comprehensive study addressing one of the enigmas of "Ribosomal and post-translational modified peptide (RiPP)" research and therefore will be of great interest for scientists interested in computational genome mining for BGCs and also experimentalists working on elucidating RiPP biosynthesis pathways and their application.

The paper is well written and the authors managed to nicely present the huge amount of methods and data ranging from in silico analyses, protein expression/purification and in vitro assays, lanthipeptide structure elucidation (MS and NMR-based) transcriptomics/co-expression analysis (on existing data), mutagenesis / in vivo studies, molecular modeling. The data is analysed and interpreted to a high standard and combined to provide a strong support for the main conclusions of the manuscript.

There are a few issues, which should be addressed before publication:

* 1. Title: The title suggests that the author's correlation-networking method is generally applicable to identify genes encoding essential biosynthetic enzymes that are not encoded within the BGCs. The authors impressively and convincingly demonstrate its use in identifying "orphan" proteases involved in RiPP biosynthesis pathways. However, in this case the target space of genes/enzymes to search for is very limited - it is known that a protease needs to be involved. And even with this prior knowledge, the search space included more than 23 million protease sequences. Is this really applicable in cases where this prior knowledge is not existing (like in the majority of cases for novel BGCs with exception of the RiPPs)? To justify the broad title, I would expect further examples (for example on the other examples of BGCs/pathways that involve "cryptic" enzymes that are mentioned in the introduction). If this is not possible I would recommend to modify the title for the application in finding orphan proteases also in the title.

Response 1: We agree with this comment and changed the title of this manuscript to “**Correlational networking guides the discovery of unclustered lanthipeptide protease-encoding genes**” (Page 1, lines 1-2). In the meanwhile, we are continuing developing our analysis and are expanding it to other types of natural products.

* 2. To reproduce the in silico workflow, the scripts used for analysing the data should be made available.

Response 2: The scripts used to construct the correlation network were all uploaded to github with a link provided in the revised manuscript (Page 23, lines 860-862).

* 3. Legend Figure 2 panel B: The authors should mention the source (species, accession) of this gene clusters, as otherwise in my opinion specifying the exact distance between BGC and protease makes not much sense.

Response 3: Thanks for the reminder. In the legend of Fig. 2 (Page 6), we added the strain name *P. polymyxa* ATCC842. Also, we added RefSeq accession for this strain in Supplementary Table 4. In addition, for all other lanthipeptide producing strains used in this manuscript, RefSeq accession numbers were added in Supplementary Table 4.

* 4. Page 10 (Bcn-gP1/2 // PttP1/2 characterisation): The authors state that both subunits need to be present for efficient proteolysis - however the HPLC diagrams show that there are also correct products formed with only subunit 2 (Bcn-gP2 or PttP2) respectively - this should be better explained.

Response 4: In the revised manuscript, we changed the corresponding part to “while excluding Bcn-gP1 generated a detectable production presumably due to Bcn-gP2 possessing catalytic residues, Bcn-gP1 and Bcn-gP2 were simultaneously required to generate the highest yield of the product...” (Page 11, lines 329-330). In addition, on Page 10, lines 313-315, it is explained that subunit 1 (Bcn-gP1 or PttP1) “contains an R/Y pair in the C-terminal domain to facilitate substrate binding” while subunit 2 (Bcn-gP2 or PttP2) “possesses an HXXEH motif essential for Zn²⁺ binding and catalytic activity”. This provides more information on why subunit 2 alone also generated detectable product, presumably because of its Zn²⁺ binding and catalytic residues, while subunit 1 facilitates substrate binding. In the meanwhile, this also helps explain why the highest proteolytic efficiency requires the pair of subunits 1 and 2.

* 5. Page 11 / Figure 5: I'm surprised that PttP1/2 show higher activity towards BcnA1 than the "native" protease Bcn-gP1/2

Response 5: We were also a bit surprised about this result at the beginning. However, after a careful consideration and based on the efficiency assay results and phylogenetic analysis, our hypothesis was that “certain proteases with general functions in the genome might have evolved more or less specific activity against class III lanthipeptides, and extra copies of them might have further evolved into pathway-specific proteases for enhanced activity and specificity” (Page 12, lines 387-390). Since the precursors BcnA1 and PttA1 both belong to the Pre_24 group and share similar sequences, it seems to be reasonable that the pathway-specific protease pair PttP1/PttP2 with higher activity for PttA1 may also have higher activity for the similar substrate BcnA1.

* 6. Is there any information available on the bioactivities of bacinapetins and paenithiopeptins?

Response 6: The similar comment was also mentioned by Reviewer #1. We totally agree that the biological activities are important. Thus, we indeed have been studying the biological activities for not only the isolated compounds of this manuscript but also other structurally related compounds. Due to the theme and the already extensive data of this manuscript, we think that it would be the best to report the biological activities in a separate paper in a timely manner, which we hope the reviewer agree with.

REVIEWERS' COMMENTS

Reviewer #1 (Remarks to the Author):

The authors have satisfactorily addressed the comments and the manuscript is now suitable for publication.

Reviewer #3 (Remarks to the Author):

In their revision, the authors have carefully addressed my previous comments and improved their manuscript accordingly. They now also have included a link to their Github repository hosting the scripts used in the analysis. As the manuscript already presents a lot of different data, I accept their argument to report the bioactivities of the isolated compounds in a separate follow-up paper.